# Yeast GPCR signaling reflects the fraction of occupied receptors, not the number

Alan Bush[1,2,†] (ID), Gustavo Vasen[1,2,†], Andreas Constantinou[1,2,‡], Paula Dunayevich[1,2,‡], Inés Lucía Patop[1,2], Matías Blaustein[1,2] & Alejandro Colman-Lerner[1,2,*] (ID)

## Abstract

According to receptor theory, the effect of a ligand depends on the amount of agonist–receptor complex. Therefore, changes in receptor abundance should have quantitative effects. However, the response to pheromone in *Saccharomyces cerevisiae* is robust (unaltered) to increases or reductions in the abundance of the G-protein-coupled receptor (GPCR), Ste2, responding instead to the *fraction* of occupied receptor. We found experimentally that this robustness originates during G-protein activation. We developed a complete mathematical model of this step, which suggested the ability to compute fractional occupancy depends on the physical interaction between the inhibitory regulator of G-protein signaling (RGS), Sst2, and the receptor. Accordingly, replacing Sst2 by the heterologous hsRGS4, incapable of interacting with the receptor, abolished robustness. Conversely, forcing hsRGS4:Ste2 interaction restored robustness. Taken together with other results of our work, we conclude that this GPCR pathway computes fractional occupancy because ligand-bound GPCR–RGS complexes stimulate signaling while unoccupied complexes actively inhibit it. In eukaryotes, many RGSs bind to specific GPCRs, suggesting these complexes with opposing activities also detect fraction occupancy by a ratiometric measurement. Such complexes operate as *push-pull* devices, which we have recently described.

**Keywords** fraction measurement; paradoxical components; ratiometric signaling; robustness
**Subject Categories** Quantitative Biology & Dynamical Systems; Signal Transduction
**Mol Syst Biol.** (2016) 12: 898

## Introduction

The canonical receptor theory (Clark, 1933; Ariens, 1954; Stephenson, 1956; Furchgott, 1966) postulates that ligands (L) bind receptors (R) following the law of mass action to form a complex (RL),

which in turn produces the actual stimulus (S), downstream of the receptor. The strength of the produced stimulus, S, depends on the intrinsic efficacy of the agonist ε (S = ε[LR]). The downstream or physiological effect E is related to S by the cell-type-specific function *f*.

$$L + R \leftrightharpoons LR \xrightarrow{\varepsilon} S \xrightarrow{f(s)} E \tag{1}$$

This can be expressed by equation 2,

$$\frac{E}{E_m} = f(S) = f(\varepsilon[LR]) = f\left(\frac{\varepsilon[R_0][L]}{K_d + [L]}\right) \tag{2}$$

where $[R_0]$ is the initial concentration of receptors, $E_m$ is the maximal possible effect, and $K_d$ is the dissociation constant, a measure of agonist–receptor affinity. The function *f* captures, in a *black box* approach, the signal transduction from active receptors down to the final effectors. This model does not depend on time; therefore, it assumes that signaling immediately reaches steady state and that the measured effect is established after L has equilibrated with R. These assumptions greatly simplify the model, but do not capture some interesting behaviors of the system. For example, we have recently reported how fast and transient signaling before L-R equilibrium is established can allow a cell to discriminate among nearly saturating concentrations of L, which are indistinguishable at steady state (Ventura *et al*, 2014).

Despite its limitations, this formulation of receptor theory still captures our fundamental understanding of the way drugs (and ligands in general) act on cells. A core prediction is that changes in $R_0$ will necessarily produce quantitative effects, evidenced in changes in the dose-response (DoR) curve, modifying the amplitude, the $EC_{50}$ (concentration of L at which 50% of the maximal effect is obtained), or both (Fig 1A; Black & Leff, 1983). Here, we explore a mechanism that could allow cells to have a response robust (invariant) to differences in the abundance of receptors (Fig 1B).

G-protein-coupled receptors (GPCRs) comprise the largest family of integral membrane receptor proteins and they are the molecular target of many therapeutic drugs (Pierce *et al*, 2002; Overington

1 Department of Physiology, Molecular and Cellular Biology, University of Buenos Aires, Buenos Aires, Argentina
2 Institute of Physiology, Molecular Biology and Neurosciences, National Research Council (CONICET), Buenos Aires, Argentina
*Corresponding author. Tel: +54 11 4576 3368; E-mail: colman-lerner@fbmc.fcen.uba.ar
†These authors contributed equally to this work as first authors
‡These authors contributed equally to this work as second authors

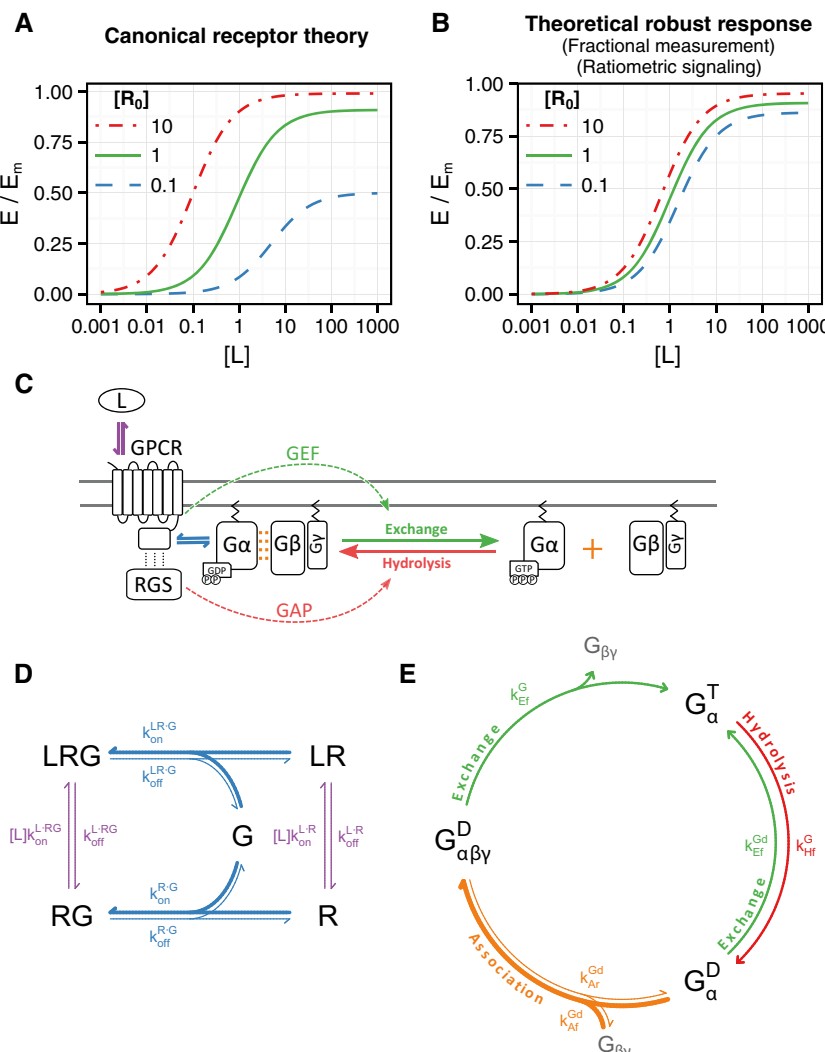

**Figure 1.  A model for G-protein-coupled receptors.**

A   Predicted dose-response curve of the normalized effect $E/E_m$ vs. ligand concentration [L], for different receptor abundances $[R_0]$. The curves were calculated according to equation 2, with $f(S) = S/(K_e + S)$ (Black & Leff, 1983), with parameters $K_e = 0.1$, $K_d = 10$, and $\varepsilon = 1$.

B   Same as in (A) for a hypothetical response robust to receptor abundance.

C   Heterotrimeric G-protein activation. The GPCR can bind to extracellular ligand (violet arrows) and couple to membrane-tethered Gα (blue arrows). Gα bound to GDP has high affinity for Gβγ (orange dots) and forms the heterotrimeric G protein. Occupied receptor acts as a GEF on Gα catalyzing the exchange of GDP for GTP (green arrows), and the subsequent dissociation between $Gα^{GTP}$ and Gβγ. Free Gα and/or Gβγ active downstream signaling components. RGS proteins act as GAPs on $Gα^{GTP}$, accelerating GTP hydrolysis (red arrows) and the subsequent association between $Gα^{GDP}$ and Gβγ. Many RGSs physically interact with GPCRs.

D   Ternary complex model. Receptor R can bind to ligand L (violet arrows) forming ligand–receptor LR, or couple to the G protein (blue arrows) forming RG complex, or can do both by forming the ternary complex LRG. Reaction rates are shown on the corresponding arrows. Note that the ligand–receptor on-rate is multiplied by the extracellular ligand concentration [L].

E   G-protein activation cycle. $G^D_{αβγ}$ denotes the heterotrimeric G protein, with the Gα subunit bound to GDP (D superscript). Exchange of GDP for GTP (green arrows) causes the dissociation of the G protein and the release of free Gβγ. Gα bound to GTP ($G^T_α$) can hydrolyze its bound nucleotide (red arrows) resulting in $G^D_α$, which can either exchange its guanine nucleotide or reassociate with free Gβγ (orange arrows). Reaction rates shown on the corresponding arrows.

*et al*, 2006). GPCRs couple to heterotrimeric guanosine nucleotide binding proteins (G proteins) composed of the α, β, and γ subunits. During signaling, this trimer undergoes cycles of dissociation and reassociation (Fig 1C). GDP-bound Gα ($Gα^{GDP}$) has high affinity for Gβγ and therefore forms the Gαβγ heterotrimer. Agonist-bound GPCRs act as guanine nucleotide exchange factors (GEFs) for Gαs, accelerating the rate at which those exchange GDP for GTP (Oldham & Hamm, 2008). $Gα^{GTP}$ dissociates from Gβγ, and both $Gα^{GTP}$ and

Gβγ, depending on the system, regulate the activity of downstream effectors (Neer, 1995; Fig 1C). Note that for a GPCR, the receptor's GEF activity is the molecular counterpart of the stimulus S in equation 2. The Gα subunit of the G protein can hydrolyze the γ-phosphate of its bound GTP, resulting in the formation of a $Gα^{GDP}$. This reaction is rather slow *per se*, but is accelerated by regulators of G-protein signaling (RGSs; Berman *et al*, 1996; Hunt *et al*, 1996; Watson *et al*, 1996; Apanovitch *et al*, 1998). Many RGS proteins

physically interact with GPCRs, either by direct binding or mediated by different types of adaptor proteins (Neitzel & Hepler, 2006). This interaction has been proposed to give specificity to RGS regulation, by localizing these proteins in the vicinity of the G proteins they regulate. Here, we also note that an RGS that interacts with a ligand-occupied GPCR forms a molecular complex with antagonistic activities: a GEF activity that activates G proteins and GAP activity that inactivates them.

The first mathematical models that incorporated coupling between receptors and intracellular components, the so-called ternary complex models (TCMs), were based on the mobile receptor hypothesis postulated by Cuatrecasas (Jacobs & Cuatrecasas, 1976; DeLean *et al*, 1980). In this model, a receptor R can bind a ligand L to form LR, couple with a second membrane component, later called G protein, to form RG, or do both, forming the ternary complex LRG (Fig 1D). The distinguishing assumptions of these models are as follows: (i) L has a higher affinity for RG than for R alone, which allowed them to explain the different apparent affinities observed for some ligands, and (ii) it is the ternary complex LRG that activates downstream effectors (DeLean *et al*, 1980). Extensions of the TCM that incorporated active and inactive states of the receptor resulted in the cubic ternary complex (CTC) model, which elegantly explains the concept of efficacy (de Haen, 1976; Samama *et al*, 1993; Weiss *et al*, 1996). Both the TCM and the CTC model are examples of thermodynamically complete models in which all possible transitions between species are considered as reversible reactions. In these models, no relevant reactions are omitted and thermodynamic restrictions such as micro-reversibility are fulfilled (that requires that if there are two reversible routes from one species to another, such as that depicted in Fig 1D, the equilibrium constants must be the same; Wyman, 1975). Later models of GPCR signaling systems incorporated the different possible states of the G protein and the transitions between them (Biddlecome *et al*, 1996; Shea *et al*, 2000; Turcotte *et al*, 2008). Some models (notably those of the yeast pheromone response system; see below) considered ligand-bound receptors as catalytic activators of the G proteins, without explicitly taking into account RG complexes (Hao *et al*, 2003; Yi *et al*, 2003; Yildirim *et al*, 2004) (Fig 1E). The TCM and the *catalytic* models may be viewed as limit cases of a more general model (Roberts & Waelbroeck, 2004).

The mating pheromone response system (PRS) of the yeast *Saccharomyces cerevisiae* is one of the best-understood GPCR signal transduction systems (Bardwell, 2005). Haploid yeast cells of mating type a (MATa) express Ste2$^{GPCR}$ which binds the peptide pheromone α-factor secreted by cells of the opposite mating type (MATα). Upon ligand binding, active Ste2$^{GPCR}$ causes the dissociation of the Ste4$^{Gβ}$-Ste18$^{Gγ}$ dimer from Gpa1$^{Gα}$. Free Gβγ recruits Ste5 to the plasma membrane, a scaffold protein that binds the components of a MAP kinase cascade. Membrane localization of Ste5 places its bound kinases in the proximity of membrane-associated Ste20$^{PAK}$ kinase, starting a phosphorylation cascade that leads to the activation of Fus3$^{MAPK}$ and Kss1$^{MAPK}$, which in turn phosphorylate downstream targets. Activation of the PRS induces cell cycle arrest, chemotropic growth toward the pheromone source, and changes in gene expression, which prepare the cells for mating.

Sst2$^{RGS}$ was the first RGS family protein to be described (Dohlman *et al*, 1995; Apanovitch *et al*, 1998), and it is one of the main negative regulators of the pheromone pathway (Chasse *et al*, 2006). It

has an N-terminal DEP-containing domain with which it interacts with the cytoplasmic C-terminal tail of Ste2$^{GPCR}$, an interaction essential for Sst2$^{RGS}$ GAP activity on Gpa1$^{Gα}$ (Ballon *et al*, 2006).

Quantitative measurements at different steps in the PRS, including receptor occupancy, G-protein dissociation, induction of transcriptional reporters, and cell cycle arrest, show good dose-response alignment (DoRA) (Yi *et al*, 2003; Yu *et al*, 2008); the EC$_{50}$ of all these activation steps is almost the same. Stated differently, the transfer function between these steps is approximately linear, which is surprising if one takes into account all the non-linear interactions in the pathway, suggesting the existence of mechanism(s) that ensure DoRA (Brent, 2009). In this regard, Fus3$^{MAPK}$ activity is required to maintain the dose response (DoR) at the Fus3 phosphorylation step (Yu *et al*, 2008). Fus3$^{MAPK}$ also exerts negative feedback on upstream Ste5 membrane recruitment, a regulatory step that could be relevant to maintain DoRA. However, inhibition of Fus3$^{MAPK}$ activity does not change the EC$_{50}$ of Ste5's membrane recruitment (Bush & Colman-Lerner, 2013), suggesting the existence of some other mechanism upstream of Ste5 that aligns input and response.

Interestingly, the quantitative response to pheromone is robust to changes in receptor abundance. Tenfold overexpression of the receptor results in a variation in mating efficiency of < 30% (Blumer *et al*, 1988; Konopka & Jenness, 1991). Overexpression of Ste2$^{GPCR}$ between 6- and 15-fold has negligible effects on cell cycle arrest (Blumer *et al*, 1988; Konopka *et al*, 1988; Konopka & Jenness, 1991; Shah & Marsh, 1996), cells that express only 10% of the normal receptor abundance have WT sensitivity, and only cells that express < 5% of the wild-type numbers of Ste2$^{GPCR}$ show decreased sensitivity to pheromone-induced cell cycle arrest (Shah & Marsh, 1996). Similarly, quantitative DoR curves of P$_{FUS1}$-lacZ reporter gene show that twofold overexpression of Ste2$^{GPCR}$ produces a small (13% reduction in the amplitude; Leavitt *et al*, 1999) or no modification of the response (Hao *et al*, 2003). More interestingly, cells with only 20% of the normal receptor level produce P$_{FUS1}$-lacZ DoR curves nearly identical to WT cells (Gehret *et al*, 2012). Furthermore, during the first 20 min of the response, the available receptors at the plasma membrane drop to around half of the original amount and then slowly increase, reaching (and then surpassing) the original levels 1 h later (Jenness & Spatrick, 1986). During this time frame, the transcription rate of a pheromone-responsive reporter gene remains virtually constant (Colman-Lerner *et al*, 2005). Taken together, these observations show that contrary to what we expect based on the receptor theory, large up- or downward changes in receptor abundance have little effect on the yeast pheromone response (Fig 1B).

In general, biological systems tend to be robust, in the sense that they maintain a rather constant performance in the face of internal and external sources of variation (Kitano, 2004; Stelling *et al*, 2004). There are a number of mechanisms that bring about such robustness, including redundancy (as in DNA repair), partially overlapping functions of two or more molecular components, modularity in the interactions among components, and feedback control (Stelling *et al*, 2004). One mechanism of particular interest involves components that catalyze antagonistic reactions (sometimes called paradoxical or *push-pull* components; Hart & Alon, 2013; Andrews *et al*, 2016), such as two-component signaling systems of bacteria (e.g., EnvZ/OmpR of *Escherichia coli* or DesK/DesR of *Bacillus subtilis*;

Albanesi *et al*, 2004; Russo & Silhavy, 1991). In many of these phosphorelay systems, the sensor protein either phosphorylates or dephosphorylates the response regulator, depending on whether the sensor itself is phosphorylated or not. Thus, there is no inactive state for the sensor: It either stimulates (*pushes*) or inhibits (*pulls*) the regulator. Because they can produce robust input–output relationships (Russo & Silhavy, 1993; Shinar *et al*, 2007; Hart & Alon, 2013; Andrews *et al*, 2016), push-pull components are of particular interest here. Recently, we showed that push-pull topologies are especially suited for bringing about DoRA in signaling pathways in general and in the PRS in particular (Andrews *et al*, 2016).

If downstream cellular responses to pheromone are robust to variation in the number of Ste2$^{GPCR}$ receptors, it follows that the PRS might respond to the *fraction*, and not the *number,* of occupied receptors (Fig 1B). One way for a cell to compute fractional receptor occupancy so as to distinguish full occupancy of 1,000 receptors from 50% occupancy of 2,000 total receptors is for occupied receptors to promote signaling and unoccupied receptors to actively inhibit it (Brent, 2009). Supporting this hypothesis, in the absence of α-factor, WT receptors suppress PRS activity induced by constitutively active receptor mutants (Konopka *et al*, 1996; Stefan *et al*, 1998). Similarly, Ste2$^{GPCR}$ mutants that are unable to bind α-factor diminish pheromone-induced PRS activity by co-expressed WT receptors (i.e., they act as dominant-negative, DN; Dosil *et al*, 1998, 2000; Leavitt *et al*, 1999; Sommers *et al*, 2000; Gehret *et al*, 2012). This inhibitory activity seems to require the unbound conformation of Ste2$^{GPCR}$, since other Ste2$^{GPCR}$ mutants that undergo normal ligand-induced conformational changes but are inactive due to impaired G-protein activation (Büküşoğlu & Jenness, 1996), do not inhibit signaling (Stefan *et al*, 1998; Dosil *et al*, 2000).

In this work, we studied the mechanism that allows the system to respond to the fraction of occupied receptors, independent of their absolute abundance. We refer to this property as "robustness to changes in receptor number", or just "robustness". We elaborated a complete mathematical model of the interaction between the receptor and the G protein. Analysis of the model showed that for parameter values consistent with the published kinetic rates and protein abundances of the PRS, the activity of the GPCR system depends on the *fraction* of occupied receptors. One of the predictions of the model was that physical interaction between the RGS and the receptor is critical for the system to respond to fractional occupancy. We tested this prediction experimentally by replacing the endogenous *SST2$^{RGS}$* with hsRGS4, a human ortholog RGS that does not interact with the receptor. This genetic perturbation eliminated the robustness to changes in receptor abundance. Conversely, forcing Ste2$^{GPCR}$ to interact with hsRGS4 by directly fusing these two proteins, or fusing the RGS domain of hsRGS4 to the DEP-containing domain of Sst2$^{RGS}$, which binds to Ste2$^{GPCR}$, restored robustness.

## Results

### Robustness depends on events upstream of Ste5 membrane recruitment

The reported robustness of the PRS to changes in the abundance of receptors (Blumer *et al*, 1988; Konopka *et al*, 1988; Reneke

*et al*, 1988; Shah & Marsh, 1996; Leavitt *et al*, 1999; Gehret *et al*, 2012) could conceivably involve various steps in the signaling cascade from receptor binding to induction of gene expression. Thus, to better determine the steps at which the mechanism that generates robustness operates, we first measured the effect of changes in receptor number on membrane recruitment of the Ste5 scaffold, the step that follows G-protein dissociation.

There are only around 500 Ste5 molecules per cell (Thomson *et al*, 2011). In order to measure relocalization of this scaffold protein, we used strains with three genomic integrations of STE5 tagged with three YFPs in tandem, under control of its endogenous promoter (3x *P$_{STE5}$-STE5-YFPx3*). To modify the abundance of *STE2$^{GPCR}$*, we replaced the *STE2$^{GPCR}$* promoter with the galactose-inducible *GAL1* promoter (*P$_{GAL1}$-STE2$^{GPCR}$*). Such strains do not respond to pheromone in glucose medium (SC-Glu), but they do in medium with galactose and raffinose (SC-Gal/Raff) (Fig EV1A). Using fluorescent α-factor (Bajaj *et al*, 2004; Toshima *et al*, 2006), we measured the Ste2$^{GPCR}$ abundance at the plasma membrane in these strains grown in SC-Gal/Raff over time after addition of low (5 nM) or high (50 nM) α-factor (Figs 2A and EV1C). Initial receptor abundance in *P$_{GAL1}$-STE2$^{GPCR}$* strains grown in SC-Gal/Raff was 5.3 ± 0.6 times greater than the value of *P$_{STE2}$-STE2$^{GPCR}$* (WT) cells grown in the same conditions (Fig EV1B). Of note, WT receptor abundance in this medium was one-third of the abundance in SC-glucose (Fig EV1B), while other components of the pathway remained fairly constant (Appendix Fig S1). After stimulation, this difference slowly disappeared, due to the combined effect of receptor endocytosis and α-factor-induced synthesis of Ste2$^{GPCR}$. Thus, this strategy of receptor overexpression was only useful for the first 15 min, enough to assess its effects on Ste5 membrane recruitment (measured during the first few minutes), but not suitable for longer-term measurements.

We measured pheromone-stimulated Ste5 recruitment at 45-s intervals for 6 min in WT *STE2$^{GPCR}$* or with *P$_{GAL1}$-STE2$^{GPCR}$*, both grown in SC-Gal/Raff. We observed no difference in the dynamics and DoR of Ste5 membrane recruitment between the two strains (Fig 2B and C). This result thus suggests that the robustness/fractional occupancy measurement mechanism operates upstream of Ste5 recruitment, perhaps at the step that couples receptor to G-protein activation.

### More than one robustness mechanism revealed by transcriptional reporters

Next, we assayed robustness at the transcriptional level, testing a wide range of Ste2$^{GPCR}$ abundances. For these experiments, we used reporter strains with YFP controlled by the pheromone-inducible *PRM1* promoter (*P$_{PRM1}$-YFP*; Colman-Lerner *et al*, 2005). In order to prevent dilution of the reporter YFP caused by cell proliferation and the cell cycle-dependent inhibition of the PRS in S-phase committed cells, these strains also contained an adenine-analogue-sensitive allele of the cyclin-dependent kinase, *cdc28-F88A* (Bishop & Shokat, 1999; Colman-Lerner *et al*, 2005). We modified Ste2$^{GPCR}$ abundance using three different approaches (Fig 2D) that resulted in sustained differences in receptor abundance, suitable for longer-term measurements. In each case, we stimulated yeast with various α-factor concentrations for two hours in the presence of 10 μM of the

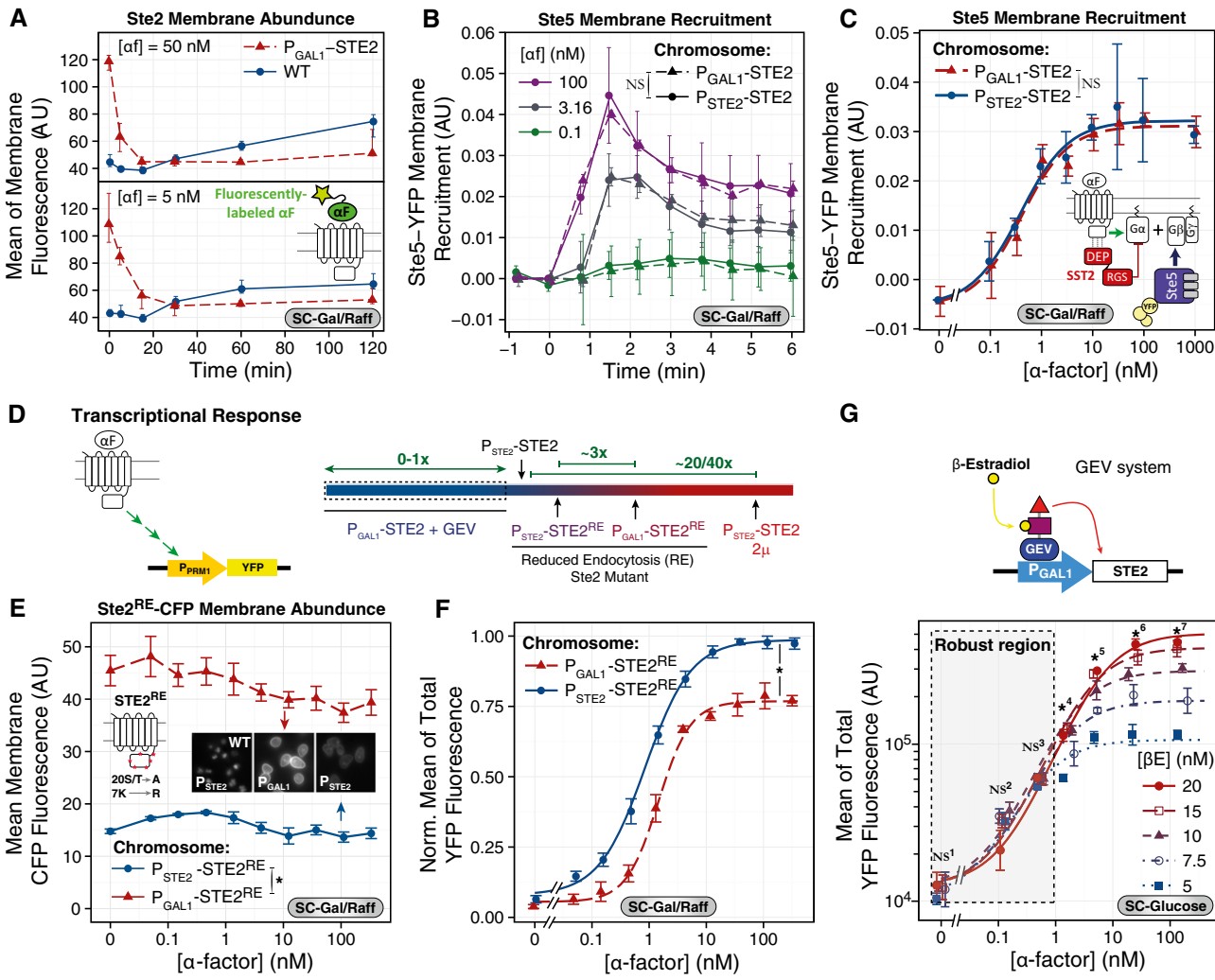

**Figure 2. The PRS is robust to changes in receptor abundance.**

A    Time course of receptor abundance. Comparison between $P_{STE2}$-STE2 and $P_{GAL1}$-STE2. We grew WT (TCY3154, blue circles) and $P_{GAL1}$-STE2 (YAB3930, red triangles) strains in SC-Gal/Raff and stimulated them with 5 or 50 nM α-factor (αF). At the indicated times, we measured receptor abundance at the membrane with fluorescent α-factor (see Appendix). Each data point represents the mean fluorescence associated with the membrane (AU) of a representative experiment; error bars represent 95% CI obtained by bootstrapping. $N > 200$ cells per data point.

B, C    Membrane recruitment of Ste5-YFPx3 is robust. We grew WT (YPP3662, solid lines, circles) and $P_{GAL1}$-STE2 (YAB5372, dashed line, triangles) yeast with YFP-tagged Ste5 in SC-Gal/Raff, stimulated them with α-factor, and imaged in the fluorescence microscope for 6 min. Plots show time courses (B) or dose responses (C) at the indicated α-factor concentrations 2.2 min post-stimulation. Data correspond to the mean increase in membrane recruitment (see Appendix and Bush & Colman-Lerner, 2013). Points represent the mean of three biological replicates, each with ~100 cells. Error bars show the 95% CI of the mean. In (B), we tested statistical significance by ANOVA for time points > 1 min: $P_{GAL1}$-STE2 vs. $P_{STE2}$-STE2, $P = 0.930$, not significant (NS). In (C), we compared coefficients obtained from non-linear mixed-effects fit to a Hill-function model. For amplitude, $P_{GAL1}$-STE2: ($0.032 \pm 0.002$) AU vs. $P_{STE2}$-STE2 ($0.031 \pm 0.002$) AU, $P = 0.690$, NS.

D–G    Transcriptional reporters. (D) Diagram shows the range of receptor abundance tested using a transcriptional reporter. We used strains containing the PRS-inducible $P_{PRM1}$-YFP reporter (Colman-Lerner *et al*, 2005). Diagram shows the ranges covered by the three strategies used: overexpression with a 2μ plasmid (Fig EV2), or with a reduced endocytosis Ste2 mutant (Ste2[RE]) or underexpression with the GEV system. (E) Ste2 membrane abundance in strains with $P_{STE2}$-STE2[RE]-CFP (ACY5563, blue circles) or $P_{GAL1}$-STE2[RE]-CFP (ACY5580, red triangles). STE2[RE]-CFP (ST20A-K7R mutant fused to CFP) has reduced endocytosis (see inset) (see also Fig EV1E). We grew cells in SC-Gal/Raff and stimulated them with the indicated concentrations of α-factor in the presence of 10 μM 1NM-PP1 (to block Cdc28-as2) for 2 h. (F) In the same cells as in (E), we measured the accumulated reporter YFP. (G) We used strain YIP5581 with $P_{GAL1}$-STE2 controlled by the GEV system (a tripartite chimera of the Gal4 DNA binding domain, the β-estradiol binding domain of the estrogen receptor, and the transactivation domain the herpes virus VP16 protein—the oval, rectangle, and triangle in the diagram, respectively; Mattioni *et al*, 1994). We grew yeast in SC-glucose, added the indicated β-estradiol concentration three hours before stimulation to induce different abundances of Ste2 (see Fig EV2F and G), and then stimulated and measured as in (F) (see Appendix). (E–G) Each data point is the mean ± SEM of three independent biological replicates. Inset in (E) shows images in the CFP channel (WT Ste2-CFP is included for comparison). In (E), statistical significance was determined by ANOVA: $P_{GAL1}$-STE2[RE] vs. $P_{STE2}$-STE2[RE], $P < 10^{-4}$ (*). In (F), we compared the coefficients obtained from non-linear mixed-effects fit to a Hill-function model. For amplitude, $P_{GAL1}$-STE2[RE]: ($0.71 \pm 0.03$) AU vs. $P_{STE2}$-STE2[RE] ($0.90 \pm 0.02$) AU, $P < 10^{-4}$ (*). In (G), we performed ANOVA at each pheromone concentration. The dashed rectangle highlights the region with $P > 0.05$, where the system is robust to changes in Ste2 abundance (βE). *P*-values: NS[1], 0.371; NS[2], 0.091; NS[3], 0.238; *[4], 0.021; *[5], 0.035; *[6], 0.005; *[7], 0.009. NS stands for not significant, and * stands for significant. Numbers next to NS and * correspond to the different sets of points (doses of α-factor) compared.

Cdc28-F88A inhibitor 1NM-PP1. In the first approach, we compared WT with a strain that expressed Ste2$^{GPCR}$ from a multicopy 2μ plasmid, which gave a 20- to 40-fold overexpression (Fig EV1D). This large increase in Ste2$^{GPCR}$ abundance did not cause any increase in the PRS response, showing a remarkable robustness at the transcriptional level. However, a small reduction was evidenced at all α-factor concentrations tested (Fig EV2A). This mild inhibition has been previously observed (Konopka & Jenness, 1991; Leavitt *et al*, 1999). In the second approach, we aimed at producing a milder overexpression of Ste2$^{GPCR}$. To do that, we used strains expressing a Ste2$^{GPCR}$ mutant with reduced endocytosis fused at its C-terminus to CFP, Ste2$^{RE}$-CFP (Ste2-20STA-7KR-CFP, in which 20 S and T Yck1 and Yck2 phosphorylation sites were mutated to A and the seven ubiquitylation K on Ste2$^{GPCR}$ were mutated to R; Ballon *et al*, 2006). Residual endocytosis of this mutant is mediated by a ubiquitin-independent pathway directed by its GPAFD sequence (Howard *et al*, 2002; Dores *et al*, 2010; Fig 2E). We compared strains that expressed this mutant from a P$_{STE2}$ or P$_{GAL1}$ promoter, both grown in SC-Gal/Raff medium. As determined by measuring membrane Ste2$^{GPCR}$ using fluorescent α-factor (as above) and membrane-localized CFP, this strategy resulted in a threefold overexpression relative to WT yeast (Figs 2E and EV1E). As in the case of the 2μ strategy, we observed a small reduction in reporter expression (Fig 2F). Because we did not observe inhibition at the Ste5 recruitment step (Fig 2B and C), these results suggest that yeast overexpressing Ste2$^{GPCR}$ dampen PRS-dependent transcription by a second mechanism that operates at longer times (after the first 6 min) or downstream of the Ste5 recruitment step and that this mechanism might actually overcompensate the increased Ste2$^{GPCR}$ resulting in a reduction in PRS activity.

Our third strategy involved testing the effect of underexpression of Ste2$^{GPCR}$. To do that, we used strains in which Ste2-CFP was under the control of P$_{GAL1}$ whose activity was controlled in turn by an β-estradiol-responsive Gal4 derivative, originally developed by Picard and collaborators (Mattioni *et al*, 1994). In these GEV (Gal4–estrogen binding domain–VP16) yeast, we added β-estradiol from 5 to 20 nM (concentrations above 20 nM are usually toxic; McIsaac *et al*, 2011) 3 h before stimulation with α-factor, resulting in Ste2$^{GPCR}$ abundances that ranged from WT at the high end to about 1/6 of WT, as judged from CFP expression and binding of fluorescent α-factor to the plasma membrane (Fig EV1F and G). Next, we measured the DoR of α-factor-induced YFP 2 h after stimulation. Unexpectedly, the effect of the changes in Ste2$^{GPCR}$ abundance depended on α-factor concentration (Fig 2G). At concentrations below 3 nM, the response was independent of β-estradiol concentration (i.e., robust to changes in Ste2$^{GPCR}$ abundance), while at and above 3 nM the amplitude of the response increased with increasing concentrations of β-estradiol (i.e., not robust).

In summary, these experiments showed an invariant response to changes in GPCR abundance at the G-protein activation step, early in the pathway activation. In the longer-term transcriptional response, results were more complex. Yeast showed a robust response to overexpression, overcompensating both small and large increases in Ste2$^{GPCR}$ (see Discussion). On the other hand, yeast underexpressing Ste2$^{GPCR}$ showed a robust response at low α-factor concentrations (below the receptor $K_d$) and no robustness above it. Therefore, it seems that more than one mechanism might be in place to control robustness to changes in receptor number.

## The *carousel* model of heterotrimeric G-protein activation

The robustness displayed by the PRS to changes in the abundance of Ste2$^{GPCR}$ means that the PRS does two things: First, it makes a measurement that converts absolute extracellular ligand concentration into a signal that depends on fractional occupancy. Second, it does so in such a way as to transmit that fraction measurement linearly, thus providing the necessary input for DoRA.

To study potential mechanisms that convert an absolute extracellular concentration into a fraction, we developed a detailed model of the coupling between receptor and G protein, the signaling step at which our experiments suggest that the robustness originates. To do so, we combined the TCM (Fig 1D; DeLean *et al*, 1980) with a plausible model of the G-protein activation cycle (Fig 1E). The resulting model can be represented in a 3D scheme as shown in Fig 3A, in a geometry reminiscent of a fairground *carousel*. In this scheme, axial (up and down) reactions represent binding of the ligand L to the receptor R, radial (in and out) reactions represent the coupling of R with the G protein, and angular reactions, the progression through the three-state G-protein activation cycle. Note that in our representation of this cycle, we considered GDP/GTP exchange and the dissociation of the Gαβγ trimer as a single reaction with a rate determined by the slowest reaction, the dissociation of GDP from Gα (see Appendix). This model shares features with previous ones (Shea *et al*, 2000; Turcotte *et al*, 2008), but it includes what turned out to be a key difference: The RGS activity is localized to the receptor. Although Sst2$^{RGS}$ is not explicitly considered, its association with the receptor is captured by the model's rates (see below). Therefore, in our model the GTP hydrolysis rate of Gα-GTP is maximal when associated with the receptor.

The complete model had 12 variables (i.e., molecular species) and 38 parameters (i.e., reaction rates and abundances; Computer Model EV1). In order to make the model tractable, we made several simplifying assumptions based on the known biology of the PRS (see details in the Appendix).

The resulting *simplified carousel* model has nine variables and 13 parameters (Table 1) (Computer Model EV1). Seven of these parameters have been measured for the pheromone response of *S. cerevisiae* (in bold in Table 1). Three have been measured for other GPCR signaling systems, and we considered them as good estimates for the corresponding value in the PRS. For three parameters, there is no reported experimental estimate. In these cases, we chose physiological values that result in a reasonable behavior of the model (see Appendix for details).

The GAP activity of the Sst2$^{RGS}$ accelerates the hydrolysis rate of Gα$^{GTP}$, increasing it more than 20-fold (Apanovitch *et al*, 1998; Yi *et al*, 2003). If this GAP activity were delocalized, the hydrolysis rate would not depend on whether Gα$^{GTP}$ is bound or not to the receptor. On the other hand, the physical association between the RGS and the receptor (Ballon *et al*, 2006; Neitzel & Hepler, 2006) suggests a localized GAP activity, resulting in a higher hydrolysis rate for receptor-coupled Gα$^{GTP}$ than for uncoupled Gα$^{GTP}$. This asymmetry in the rates can be formalized as $k_{Hf}^{LRGt} \approx k_{Hf}^{RGt} \gg k_{Hf}^{Gt}$, where $k_{Hf}^{LRGt}$ is the hydrolysis rate of Gα$^{GTP}$ coupled to ligand-occupied receptor, $k_{Hf}^{RGt}$ is the hydrolysis rate of Gα$^{GTP}$ coupled to unoccupied receptor, and $k_{Hf}^{Gt}$ is the hydrolysis rate of uncoupled Gα$^{GTP}$.

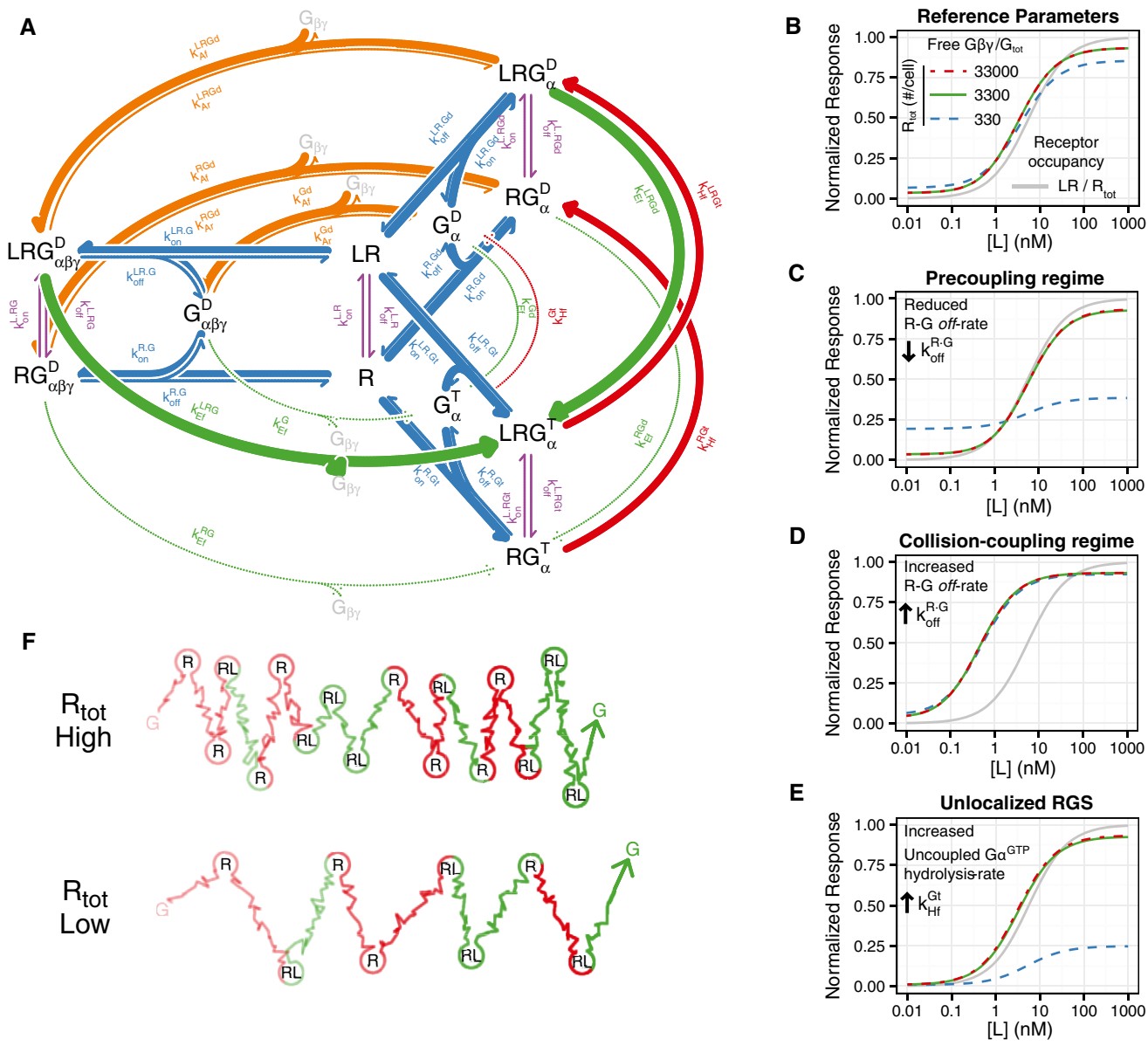

**Figure 3. The *carousel* model shows a response robust to changes in GPCR abundance.**

A   Scheme of the *carousel* model resulting from the combination of the ternary complex model (Fig 1D) and the G-protein activation cycle (Fig 1E). Axial violet arrows indicate receptor–ligand binding reactions, and radial blue arrows represent receptor–Gα coupling reactions. Angular arrows indicate progression through the G-protein activation cycle; green arrows represent the exchange of GDP for GTP in Gα and the corresponding dissociation of Gβγ, red arrows the hydrolysis of GTP by Gα, and orange arrows association between Gα$^{GDP}$ and Gβγ. Superindices D and T denote the GDP- and GTP-bound states of Gα, respectively. The width of the arrows shows the relative rates of the reactions (see Table 1).

B–E   (B) Steady-state dose-response curves, resulting from the *carousel* model and the reference parameter set (Table 1). The gray curve represents receptor occupancy normalized to total receptor. Color curves show the amount of free Gβγ normalized to the total amount of G protein, for the indicated receptor abundances. Note that the red dashed line (33,000 receptors/cell) falls over the green solid line (3,300 receptors/cell). (C) Same as in (B) for $k_{off}^{R\cdot G} = 0.001\ s^{-1}$. (D) Same as in (B) for $k_{off}^{R\cdot G} = 10\ s^{-1}$. (E) Same as in (B) for $k_{Hf}^{RGt} = k_{Hf}^{Gt} = 0.11\ s^{-1}$.

F   Schematic representation of the diffusion of a G protein in the plasma membrane and its interaction with free (R) and occupied (RL) receptors, for a total number of receptors of 16 (R$_{tot}$ High) or 8 (R$_{tot}$ High), in a case of 50% receptor occupancy. Random path lines represent diffusion through the membrane from one receptor to the next and circles, interaction times between G proteins and receptors. Red and green portions of the Gα trajectory represent GDP- and GTP-bound states, respectively. The G protein diffuses from left to right. Note the fraction of time the G protein is active is identical despite the difference in receptor abundance.

## The *carousel* model shows robustness to receptor abundance and DoRA

To determine whether the *carousel* model can compute fractional receptor occupancy, we simulated steady-state DoR curves for different levels of total receptors (Computer Model EV1). Notably, using the reference parameters (Table 1) the output of the model (free-Gβγ DoR curves) is practically unchanged by 100-fold variation (10× increase and 10× decrease) in receptor abundance (Fig 3B), indicating that indeed, the levels of free Gβγ reflect the

**Table 1. Reference parameter values for the simplified *carousel* model (highlighted in bold are the parameters measured in the pheromone response of *S. cerevisiae*).**

| Parameter | Value | References | Description |
|---|---|---|---|
| $K_d^{L \cdot R}$ | **5.6 nM** | Jenness *et al* (1986), Weiner *et al* (1993), David *et al* (1997), Dube & Konopka (1998), Dosil *et al* (2000), Lee *et al* (2001), Bajaj *et al* (2004), Kusari *et al* (2004) | $K_d$ between L and R |
| $k_{off}^{L \cdot R}$ | **0.001 s$^{-1}$** | Jenness *et al* (1983), Bajaj *et al* (2004) | LR complex off-rate |
| $K_d^{R \cdot G}$ | 33 nM | Alves *et al* (2003, 2005) | $K_d$ between R and G |
| $k_{off}^{R \cdot G}$ | 0.1 s$^{-1}$ | Hein *et al* (2006) | RG complex off-rate |
| $K_d^{Gd \cdot G\beta\gamma}$ | 0.01 nM | Estimated in this work | $K_d$ between $G_\alpha^{GDP}$ and $G_{\beta\gamma}$ |
| $k_{Af}^{Gd}$ | 3.2 nM s$^{-1}$ | Estimated in this work | Association rate of $G_\alpha^{GDP}$ with $G_{\beta\gamma}$ |
| $k_{Ef}^{G}$ | **0.00062 s$^{-1}$** | Apanovitch *et al* (1998) | Exchange rate of uncoupled $G_\alpha^{GDP}$ |
| $k_{Ef}^{RG}$ | 0.00062 s$^{-1}$ | Estimated in this work | Exchange rate of $G_\alpha^{GDP}$ coupled to unoccupied R |
| $k_{Ef}^{LRG}$ | 1.5 s$^{-1}$ | Biddlecome *et al* (1996), Mukhopadhyay & Ross (1999) | Exchange rate of $G_\alpha^{GDP}$ coupled to ligand-occupied R |
| $k_{Hf}^{Gt}$ | **0.002 s$^{-1}$** | Apanovitch *et al* (1998), Yi *et al* (2003) | Hydrolysis rate of uncoupled $G_\alpha^{GTP}$ |
| $k_{Hf}^{RGt}$ | **0.11 s$^{-1}$** | Yi *et al* (2003) | Hydrolysis rate of $G_\alpha^{GTP}$ coupled to unoccupied R |
| $R_{tot}$ | **1,400 nM** | Thomson *et al* (2011) | Total R concentration |
| $G_{tot}$ | **860 nM** | Thomson *et al* (2011) | Total G-protein concentration |

$K_d$, dissociation constant; $k_{off}$, dissociation (off-)rate; R, receptor; G, G protein; and L, ligand.

fraction of occupied receptor and not the absolute amount (the same results are found for total $G_\alpha^{GTP}$ as an output, not shown). Furthermore, these DoR curves are well aligned with receptor occupancy; that is, the information about fraction occupancy is transmitted to free $G_{\beta\gamma}$ linearly (the system shows DoRA; Yu *et al*, 2008). Therefore, without the need of fitting the model to data, the *carousel* model can qualitatively predict the two system-level behaviors of the PRS we sought to explain.

Next, we set out to analyze the *carousel* model to extract the aspects of its operation that enabled fractional occupancy measurement and DoRA. We noticed that the behavior of the system depended strongly on the dissociation rate of the receptor–G$\alpha$ complex ($k_{off}^{R \cdot G}$; Fig 3C and D). This parameter dictates the coupling mode between the receptor and the G protein (Lauffenburger & Linderman, 1993; Roberts & Waelbroeck, 2004). If the receptor takes much longer to unbind from G$\alpha$ than from the ligand ($k_{off}^{R \cdot G} \ll k_{off}^{L \cdot R}$), then the system works in the *precoupling* (or *ternary complex*) regime, in which an occupied receptor will only activate a single G$\alpha$, the one to which it is bound. In the opposite situation, if the receptor unbinds more quickly from the G$\alpha$ than from the ligand ($k_{off}^{R \cdot G} \gg k_{off}^{L \cdot R}$), an occupied receptor may interact with and activate several G$\alpha$s, acting like an enzyme. This situation corresponds to the *collision-coupling* (or *catalytic reaction*) regime (Lauffenburger & Linderman, 1993; Roberts & Waelbroeck, 2004). The value of this parameter in the reference set for the PRS lies in between these two extremes (Hein *et al*, 2006).

Hence, we studied the effect of changing dissociation rate of the receptor–G$\alpha$ complex $k_{off}^{R \cdot G}$ on the robustness to changes in the abundance of the receptor. When operating in the *precoupling regime* ($k_{off}^{R \cdot G} \ll k_{off}^{L \cdot R}$), there is perfect DoRA between receptor occupancy and G-protein activation (Fig 3C), consistent with the linear transfer function obtained if each receptor associates with and activates a single G$\alpha$. Increasing the number of receptors in the simulation had no effect (Fig 3C), as in this regime uncoupled receptors do not

affect signaling. On the other hand, reducing the number of receptors below the level of G proteins decreased the response (Fig 3C), as uncoupled G proteins cannot be activated in this regime. Thus, in the *precoupling regime* the model does not exhibit robustness to decreases in receptor abundance and therefore cannot compute fractional receptor occupancy. Note that our experimental results do show robustness between 0.3 and 1.5× WT abundance of receptor, both at the level of Ste5 membrane recruitment (Fig 2B and C) and in the transcriptional response at low doses of $\alpha$-factor (Fig 2G), indicating that the PRS does not operate in the *precoupling regime*.

In the *collision-coupling* regime ($k_{off}^{R \cdot G} > k_{off}^{L \cdot R}$), we found almost perfect robustness to receptor abundance (Fig 3B and D). In this regime, G$\alpha$ subunits interact randomly with both occupied and unoccupied receptors. An encounter with a ligand-occupied receptor will tend to leave G$\alpha$ bound to GTP (i.e., active). This is because the GDP exchange rate ($k_{Ef}^{LRGd}$) is greater than the GTP hydrolysis rate for a G$\alpha$ coupled to a ligand-occupied receptor ($k_{Hf}^{LRGt} < k_{Ef}^{LRGd}$; Table 1). On the contrary, an encounter with an unoccupied receptor will tend to leave G$\alpha$ in its GDP-bound state (i.e., inactive), because in this case the hydrolysis rate is greater than the exchange rate ($k_{Hf}^{RGt} > k_{Ef}^{RGd}$; Table 1). Consequently, in the *collision-coupling* regime occupied receptors tend to activate all the G$\alpha$ subunits with which they interact, while unoccupied receptors tend to inactivate them. This constitutes a *ratiometric* mechanism by which G protein can report the *fraction* of occupied receptor, since by responding to both occupied and unoccupied receptors, an increase in the absolute abundance of receptor will increase both activating and inactivating activities.

**In a fraction measurement *regime*, the state of a G$\alpha$ subunit is determined by the occupancy state of the last receptor it interacted with**

Within the *collision-coupling* regime, the system can exhibit a free-G$_{\beta\gamma}$ DoR curve either well aligned (DoRA) or more sensitive (lower

$EC_{50}$) than the receptor's occupancy curve (Fig 3B and D, respectively). The key difference is the relation between the receptor–G$\alpha$ dissociation rate and the GTP hydrolysis rate. If dissociation is faster than hydrolysis ($k_{off}^{R\text{-}G} \gg k_{Hf}^{RGt}$), the lifetime of the receptor–G-protein complex is not long enough to allow the RGS-stimulated hydrolysis of GTP to take place; therefore, most interactions between G$\alpha$ and unoccupied receptors do not change the G$\alpha$ state. In this scenario, G$\alpha$ molecules activated by few occupied receptors can accumulate, increasing the sensitivity of the response (Fig 3D). On the other hand, if the receptor–G$\alpha$ dissociation is slower than GTP hydrolysis ($k_{Hf}^{RGt} \gtrsim k_{off}^{R\text{-}G}$), there is good alignment between the DoR curves (i.e., the system exhibits DoRA; Fig 3B). In this case, the duration of the receptor–G$\alpha$ complex is long enough for both the GEF- and GAP-catalyzed reactions to occur, but short enough to allow each G$\alpha$ subunit to interact with several receptors. Therefore, the state of a G$\alpha$ subunit is essentially determined by the occupancy state of the last receptor it interacted with (Fig 3F). If G proteins randomly interact with several receptors, the fraction of active G$\alpha$ will be equivalent to the fraction of occupied receptors, and the system will show both DoRA and robustness to receptor abundance (fractional receptor occupancy measurement).

Note that for this regime to work as explained, the association between the RGS and the receptor is critical, as this allows unoccupied receptors to inactivate G proteins. To test the importance of this association for robustness, we modified the model increasing the rate of GTP hydrolysis by uncoupled G$\alpha$ to match the rate of that reaction when it is coupled to the receptor ($k_{Hf}^{RGt} = k_{Hf}^{Gt}$), thus representing a state in which the RGS does not need to be associated with the GPCR. As expected, reducing the number of receptors decreased the response (Fig 3E) due to the low GEF activity and the receptor-independent rate of hydrolysis of G$\alpha$-GTP. On the other hand, increasing the number of receptors in the simulation had little effect. This condition is actually similar to the scenario in the collision-coupling regime (Fig 3D) since, due solely to mass action, virtually all G$\alpha$ subunits are coupled to receptors at any time and free-G$\alpha$-GTP hydrolysis is negligible. This simulation showed that robustness of the PRS as a whole might critically depend on the interaction between receptor and RGS and that the effect of this interaction is more relevant when receptor abundance is lower than WT.

Due to the importance of GPCR–RGS interaction, we decided to include it explicitly in the model (Fig EV3). We found that with reasonable values for the new parameters (see Appendix), this extended *carousel* model behaves essentially in the same way as the simplified model (Fig EV3B–D) (Computer Model EV1).

Taken together, the above modeling analysis suggests that the fact that Sst2$^{RGS}$ acts on G$\alpha^{GTP}$ only when in complex with Ste2$^{GPCR}$ could be fundamental for the ability of the PRS to measure fractional receptor occupancy.

### Replacing *SST2*$^{RGS}$ by hsRGS4 eliminates robustness to changes in receptor abundance

The above work predicts that localized RGS activity is required for the system to respond to the fraction of occupied receptors. In order to test this prediction, we decided to replace the endogenous RGS, *SST2*$^{RGS}$, by the human ortholog hsRGS4. When expressed in yeast, hsRGS4 rescues the supersensitive phenotype of $\Delta sst2^{RGS}$ (Druey *et al*, 1996; Srinivasa *et al*, 1998). Important here, it localizes to the

plasma membrane using its N-terminal domain, and it has no DEP domain, the domain of Sst2$^{RGS}$ that interacts with the receptor (Ballon *et al*, 2006); therefore, we expect it to be evenly distributed over the surface of the inner plasma membrane and not localized to the receptor. Indeed, expression of hsRGS4 tagged with C-terminal CFP from a constitutive promoter ($P_{ACT1}$-hsRGS4-CFP) resulted in homogeneous CFP signal on the periphery of the cell, even in the absence of Ste2$^{GPCR}$ (Fig EV4A). We therefore expected $\Delta sst2^{RGS}$ cells expressing hsRGS4 to have an RGS activity not restricted to the receptor, and, consequently, that those cells would fail to measure the fraction of occupied receptor and instead exhibit a response dependent on the abundance of receptors. Sst2$^{RGS}$ expression is relatively low in unstimulated cells, but it is significantly induced by $\alpha$-factor (Roberts *et al*, 2000). Thus, to obtain an activity of hsRGS4 similar in its dynamics to that of Sst2$^{RGS}$, for the following experiments we expressed it under the control of the endogenous $P_{SST2}$ by replacing the *SST2*$^{RGS}$ ORF with that of *hsRGS4-CFP* (Fig 4A).

In the first set of tests, we measured Ste5 plasma membrane recruitment dynamics for the first 6 min, at various $\alpha$-factor concentrations, using the same strains as in Fig 2 (comparing $P_{GAL1}$-*STE2*$^{GPCR}$ with $P_{STE2}$-*STE2*$^{GPCR}$, both grown in SC-Gal/Raff), but now expressing hsRGS4 instead of Sst2$^{RGS}$. In contrast to the robustness displayed by *SST2*$^{RGS}$ strains (Fig 2B and C), in yeast with *hsRGS4-CFP*, the degree of recruitment of Ste5-YFPx3 correlated with Ste2$^{GPCR}$ abundance (Fig 4B and C). This indicates that the robust response of WT cells to changes in GPCR abundance required Sst2$^{RGS}$.

To demonstrate that Sst2$^{RGS}$ interaction with Ste2$^{GPCR}$ was the reason for the robustness, and not an unrelated activity of Sst2$^{RGS}$, we tried to restore robustness by targeting hsRGS4 to the receptor. To do that, we made a strain that, instead of the endogenous RGS Sst2, expressed the N-terminal region of Sst2 (aa 1–419) containing its DEP domains fused to the C-terminal region of hsRGS4 (aa 34–205) containing its RGS domain (Tanaka & Yi, 2010), followed by CFP (Fig 4D). This strain complemented $\Delta sst2$, but less well than native hsRGS4, as judged by a halo assay (Fig EV4B). Then, we measured Ste5 recruitment, as above. In contrast to the *hsRGS4-CFP* yeast, the *DEP(SST2)-RGS(hsRGS4)-CFP* strain showed a response independent of Ste2$^{GPCR}$ abundance (compare Fig 4B and C with 4E and F). These results are consistent with the prediction that robustness to receptor abundance requires that the RGS be physically associated with the GPCR.

Next, we tested the Sst2$^{RGS}$ by hsRGS4 replacement strategy in the transcriptional reporter assay. We had observed in both overexpression strategies (2$\mu$ plasmid or the reduced endocytosis Ste2$^{GPCR}$ mutant driven by $P_{GAL1}$) that increasing Ste2$^{GPCR}$ abundance led to partial inhibition of the PRS response (Figs 2F and EV2A). Replacing *SST2*$^{RGS}$ by *hsRGS4* did not affect this inhibition (Figs 5 and EV2B), nor the PRS transcriptional response overall. This result is in agreement with our model, which predicted that robustness to increases in Ste2$^{GPCR}$ did not depend on the interaction between RGS and the GPCR (Fig 3E). The inhibition by increased GPCR abundance could also be observed in strains devoid of any Sst2$^{RGS}$ (Fig EV4D), indicating that it was unrelated to an RGS inhibitory activity (see Discussion).

In the $\beta$-estradiol-controlled GEV strains, which allowed Ste2$^{GPCR}$ abundance lower than WT, we had observed that the PRS was robust when stimulated with low $\alpha$-factor concentrations, but not with high $\alpha$-factor (Fig 2G). However, when we replaced Sst2$^{RGS}$ by

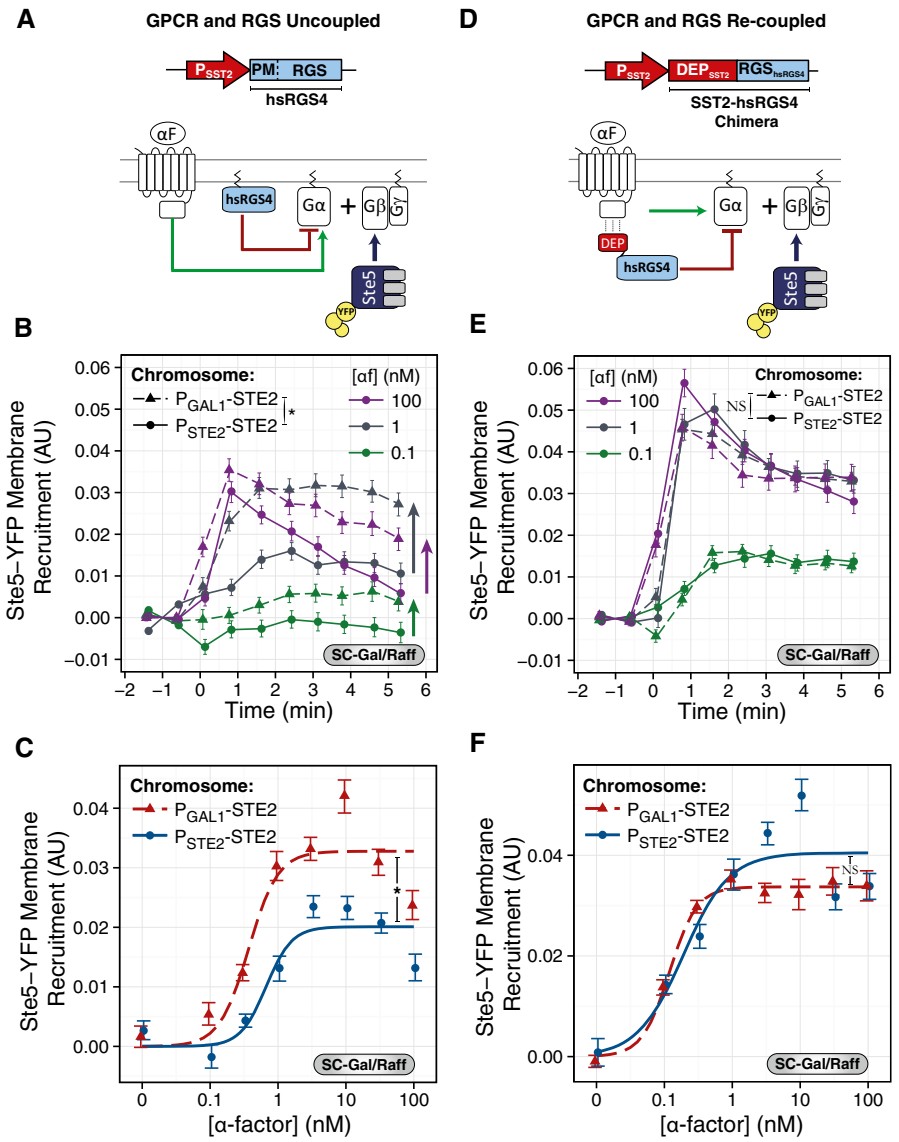

**Figure 4.   Replacing Sst2 with RGS4 abolishes robustness at the Ste5 recruitment step.**

A–C   GPCR and RGS uncoupled. (A) Scheme of the uncoupling strategy. We expressed hsRGS4-CFP under the endogenous SST2 promoter ($P_{SST2}$). hsRGS4 attaches to the plasma membrane via its N-terminal PM domain and acts on Gα via its C-terminal RGS domain (Fig EV4A–C). (B, C) We grew YFP-tagged Ste5 strains expressing $P_{SST2}$-*hsRGS4-CFP* as the only RGS with *STE2* under the control of $P_{STE2}$-*STE2* (ACY5612, circles, solid lines) or $P_{GAL1}$-*STE2* (ACY5613, triangles, dashed lines) in SC-Gal/Raff, stimulated as in Fig 2B and C, with the indicated α-factor and imaged for the first 6 min. Plots show time courses (B) or dose responses at 1.6 min (C). Data correspond to the mean ± SEM of three independent biological replicates of the increase in Ste5 membrane recruitment, as in Fig 2B and C.

D–F   GPCR and RGS recoupled. (D) Scheme of the recoupling strategy. We expressed a fusion between the *SST2* DEP domain and the hsRGS4 RGS domain under the endogenous $P_{SST2}$. This chimera binds Ste2 via its DEP domain and acts on Gα via its C-terminal RGS domain (Fig EV4B and C). (E, F) Same as (B, C) but using the DEP-RGS4 chimera strains (ACY5614 and ACY5615). Arrows highlight that the Ste2-overexpressing yeast ($P_{GAL1}$) recruit more Ste5 than WT, in contrast to SST2 yeast in Fig 2B and C or the forced-interacting chimera strains (DEP-RGS4) in (D–F). Points represent the mean ± SEM of three biological replicates, containing a total of > 200 cells.

Data information: For time courses, we tested statistical significance by ANOVA for time points > 0.5 min: $P_{GAL1}$-*STE2* vs. $P_{STE2}$-*STE2*: in (B), $P < 10^{-4}$ (*); in (E), $P = 0.836$ (NS). For dose responses, we compared the coefficients obtained from non-linear mixed-effects fit to a Hill-function model. In (C), the amplitude of $P_{GAL1}$-*STE2*: (0.019 ± 0.003) AU and $P_{STE2}$-*STE2* (0.030 ± 0.002) AU, $P = 0.004$ (*). In (F), the amplitude of $P_{GAL1}$-*STE2*: (0.044 ± 0.004) AU and $P_{STE2}$-*STE2* (0.052 ± 0.003) AU, $P = 0.058$ (NS).

hsRGS4, the PRS lost robustness at all α-factor concentrations (Fig 6A and C): For a given α-factor concentration, higher Ste2$^{GPCR}$ expression (more β-estradiol) resulted in higher reporter expression. Here, we used a different method for localizing hsRGS4 to the receptor to restore robustness. We forced the association between

Ste2$^{GPCR}$ and hsRGS4 by directly fusing the RGS domain of hsRGS4 to the C-terminus of Ste2$^{GPCR}$. In this way, we sought to bypass the dependency on the DEP domain of Sst2$^{RGS}$. Binding of Sst2$^{RGS}$ to Ste2$^{GPCR}$ via this domain may protect Ste2$^{GPCR}$ from endocytosis (Venkatapurapu *et al*, 2015). Thus, it was important to distinguish

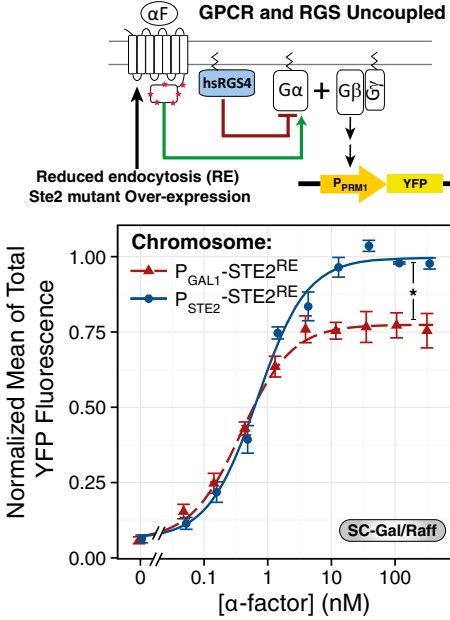

**Figure 5. Transcriptional reporters and Ste2 overexpression. RGS-independent robustness.**

As in Fig 4, we uncoupled the GPCR and RGS replacing *SST2* with *hsRGS4*, here in strains expressing the reduced endocytosis Ste2$^{RE}$ mutant (20STA-7KR) under the $P_{STE2}$ (ACY5626) or $P_{GAL1}$ (ACY5630) in SC-Gal/Raff. We added the indicated α-factor concentration for two hours in the presence of 10 μM 1NM-PP1 before measuring accumulated YFP reporter. Plot shows dose responses, and data correspond to the mean ± SEM YFP normalized to the maximum of the non-overexpressing strain of three independent experiments. Note that overexpressing strains show a reduced response relative to WT. For evaluating statistical significance, we compared the coefficients obtained from non-linear mixed-effects fit to a Hill-function model. For amplitude, $P_{GAL1}$-STE2$^{RE}$: (0.78 ± 0.03) vs. $P_{STE2}$-STE2$^{RE}$ (0.99 ± 0.02), $P < 10^{-4}$ (*).

which of the functions of Sst2$^{RGS}$ (GAP or endocytosis protection) was relevant to confer robustness to changes in receptor number. In Δ*sst2*$^{RGS}$ cells, the Ste2-hsRGS4-CFP chimera correctly localized to the plasma membrane, it was able to undergo α-factor-induced internalization and it had a similar α-factor sensitivity as WT cells (Fig EV4C, E and F). Remarkably, in yeast expressing the Ste2-hsRGS4-CFP chimera, robustness of the DoR measured by reporter expression to variation in GPCR concentration was surprisingly robust at high α-factor concentrations, a region in which the PRS in WT cells was not, and less so at low α-factor (Fig 6B and C). At low α-factor, increasing Ste2-RGS4 (higher β-estradiol doses) somewhat lowers the response. Notably, closer inspection of the simulations of the *carousel* model revealed that, as in this experiment, increasing R reduces the output in the low L region (Fig 6B inset) (see Discussion). In WT cells, Sst2$^{RGS}$ interaction with Ste2$^{GPCR}$ is under regulation, so it might not be attached to Ste2$^{GPCR}$ in all conditions (Ballon *et al*, 2006). Thus, it is perhaps not surprising that yeast with Ste2-RGS4 are a better match than WT to the *carousel* model, since in the model the RGS activity is always linked to the GPCR.

Taken together, the above results support the hypothesis that robustness to changes in Ste2$^{GPCR}$ abundance depends on a physical interaction between Ste2$^{GPCR}$ and Sst2$^{RGS}$, as our modeling effort predicted. In other words, our data suggest that a bifunctional GPCR-RGS complex is essential for fractional occupancy response.

## Replacing *SST2*$^{RGS}$ by hsRGS4 increases cell-to-cell variability in the response

Here, we found a considerable cell-to-cell variation in cell surface abundance of Ste2$^{GPCR}$, both before and after a 2-h stimulation with pheromone, as measured by fluorescent α-factor binding ($\eta^2 = 0.19 \pm 0.03$ and $\eta^2 = 0.23 \pm 0.06$; $\eta^2$ corresponds to the CV$^2$). In cells that respond to the fraction of occupied receptors, this variability should not be propagated down the signaling pathway to the measured response. We tested this idea using the GEV strains (Fig 6D). We found that yeast with *SST2*$^{RGS}$ had lower variability in the expression of the $P_{PRM1}$-YFP reporter than yeast with hsRGS4 at all α-factor concentrations tested, consistent with our hypothesis. In addition, yeast with Ste2$^{GPCR}$-RGS4-CFP fusion displayed a reduced variability, similar to *SST2*$^{RGS}$, at high α-factor concentrations. Notably, this is the same concentration range in which this strain was robust to changes in receptor abundance (Fig 6B). These results further support the role of the GPCR-RGS complex in fractional occupancy measurement.

## Dominant-negative receptors inhibit signaling by recruiting the RGS

Our data and modeling results indicate that the PRS responds to the *fraction* of occupied receptors and that the mechanism of fraction measurement involves inhibition of signaling by unbound Ste2$^{GPCR}$-Sst2$^{RGS}$ complexes. Mutant receptors that do not bind ligand (DN receptors) provide the opportunity of directly altering the fraction of occupied (active) receptors to test our idea. For example, co-expression of equal amounts of WT and DN receptors would result in 50% receptor occupancy when exposed to saturating concentrations of ligand. If that system responds to fractional occupancy, then it should exhibit 50% of its maximal response.

Here, we used one such mutant receptor, Ste2-F204S (Dosil *et al*, 1998). We expressed it from a single-copy CEN plasmid in STE2$^{GPCR}$ or Δ*ste2*$^{GPCR}$ strains and measured the accumulation of the transcriptional reporter after stimulation with α-factor. Consistent with an extremely reduced affinity of Ste2-F204S for pheromone, cells expressing just this receptor showed only residual PRS activity at high α-factor (Fig EV5A). In cells co-expressing both receptor variants, expression of the DN Ste2$^{GPCR}$ did not affect the abundance of membrane-localized, endogenous Ste2$^{GPCR}$ (Fig EV5B; Dosil *et al*, 1998). The DoR of the DN Ste2$^{GPCR}$-expressing cells showed reduced amplitude (Fig 7A), confirming it acts as DN in our system. The maximum response was just above 60% WT, close to the theoretical 50% reduction expected from our proposed mechanism. This small difference might be accounted for by the residual activity displayed by the DN Ste2$^{GPCR}$ (Fig EV5A).

Notably, the Ste2-F204S receptor was unable to inhibit reporter expression in strains that expressed hsRGS4-CFP instead of Sst2$^{RGS}$ (Fig 7B), consistent with our hypothesis that DN receptors depended on their ability to bind an RGS for their inhibitory role. To test that notion further, we fused the RGS domain of hsRGS4 to the C-terminus of the DN receptor and asked whether this fusion (Ste2-F204S-RGS4-CFP) could inhibit the PRS response activated by WT receptors. Indeed, strains that expressed this chimera as the only source of RGS activity exhibited a rather normal α-factor response, compared to the supersensitivity of the control without RGS (Fig 7C),

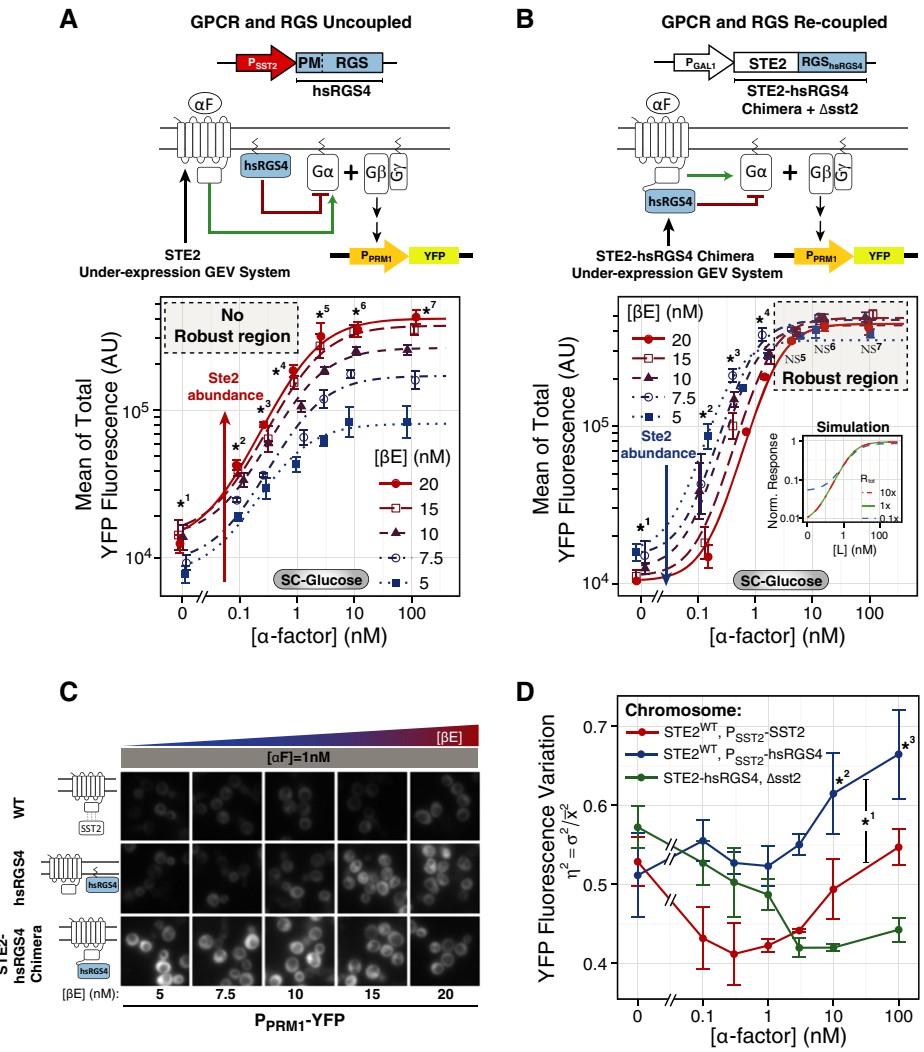

**Figure 6.  Transcriptional reporters and Ste2 underexpression. GPCR–RGS interaction-dependent robustness.**

A   GPCR and RGS uncoupled. As in Fig 4, we uncoupled the GPCR and the RGS replacing SST2 with RGS4. We grew a $P_{PRM1}$-YFP strain with the GEV system, $P_{GAL1}$-STE2 and $P_{SST2}$-hsRGS4-CFP (YGV5642) in SC-glucose. Three hours before stimulation with the indicated α-factor concentration and 10 μM 1NM-PP1, we added the indicated β-estradiol concentration. Two hours later, we imaged to measure the accumulated YFP reporter. Plot shows the mean ± SEM of YFP vs. α-factor of three independent experiments.

B   GPCR and RGS recoupled. We recoupled Ste2 and the RGS4 by fusing the RGS domain of RGS4 to the C-terminus of Ste2. Data are as in (A) except that we used a $P_{PRM1}$-YFP strain with the GEV system, $P_{GAL1}$-Ste2-hsRGS4-CFP Δsst2 (YGV5620). Arrows highlight the effect of increasing Ste2. The dashed rectangle in (B) highlights the region in which the system is robust to changes in Ste2 abundance (βE). Inset shows the same simulation shown in Fig 3B but plotted in log–log scale, so as to better compare with data in this panel. Plot shows the mean ± SEM of YFP vs. α-factor of three independent experiments.

C   YFP channel images of $P_{PRM1}$-YFP strains with GPCR–RGS natively coupled via Sst2 (top, YIP5581), uncoupled ($P_{SST2}$-hsRGS4-CFP, middle, YGV5642) or recoupled ($P_{GAL1}$-Ste2-hsRGS4-CFP, YGV5620), pre-incubated for 3 h with increasing concentrations of β-estradiol, and then stimulated as above for 2 h with 1 nM α-factor and 10 μM 1NM-PP1. Note the lost robustness (increased YFP as a function of βE) in the uncoupled strain and the recovered robustness in the recoupled strain.

D   Uncoupling the GPCR from the RGS increases variability in the PRS. Plot shows cell-to-cell variability in $P_{PRM1}$-YFP reporter accumulation ($\eta^2_{(YFP)}$ = STD$^2$/mean$^2$ ± SEM) using the data from (A, B), and from Fig 2G of cultures pre-incubated with 5 nM β-estradiol and then stimulated with indicated α-factor concentrations. Data are from three independent experiments.

Data information: (A, B) For statistical comparisons, we performed ANOVA at each pheromone concentration. In (B), the dashed rectangle highlights the region with $P > 0.05$, where the system is robust to changes in Ste2 abundance (βE). In (A), P-values: *[1], 0.0414; *[2], 0.0003; *[3], 0.0091; *[4], 0.0170; *[5], 0.0250; *[6], 0.0006; *[7], 0.0041. In (B), P-values: *[1], 0.0361; *[2], 0.0054; *[3], 0.0116; *[4], 0.0497; NS[5], 0.1265; NS[6], 0.1610; NS[7], 0.05. (C, D) Statistical analysis was performed by ANOVA. Strains expressing Sst2 were less variable than those expressing hsRGS4 at all α-factor concentrations ($P = 6.3 \times 10^{-5}$, *[1]). To compare strains with hsRGS4 and STE2-hsRGS4, given the positive interaction of strain with pheromone dose, we used a Tukey post-test. At 10 nM, $P = 0.039$(*[2]); and at 100 nM, $P = 0.011$(*[3]). P-values in each case were obtained by ANOVA. NS stands for not significant, and * stands for significant. Numbers next to NS and * correspond to the different sets of points (doses of α-factor) compared.

consistent with the idea that DN receptors need to bind RGS to be inhibitory. Interestingly, in the reciprocal experiment [i.e., co-expression of the DN receptor with the WT receptor fused to RGS4

(Ste2-hsRGS4-CFP)], in which the DN receptor cannot recruit an RGS to the membrane, there was a small but significant inhibition of signaling (Fig EV5C), suggesting that there might be another parallel

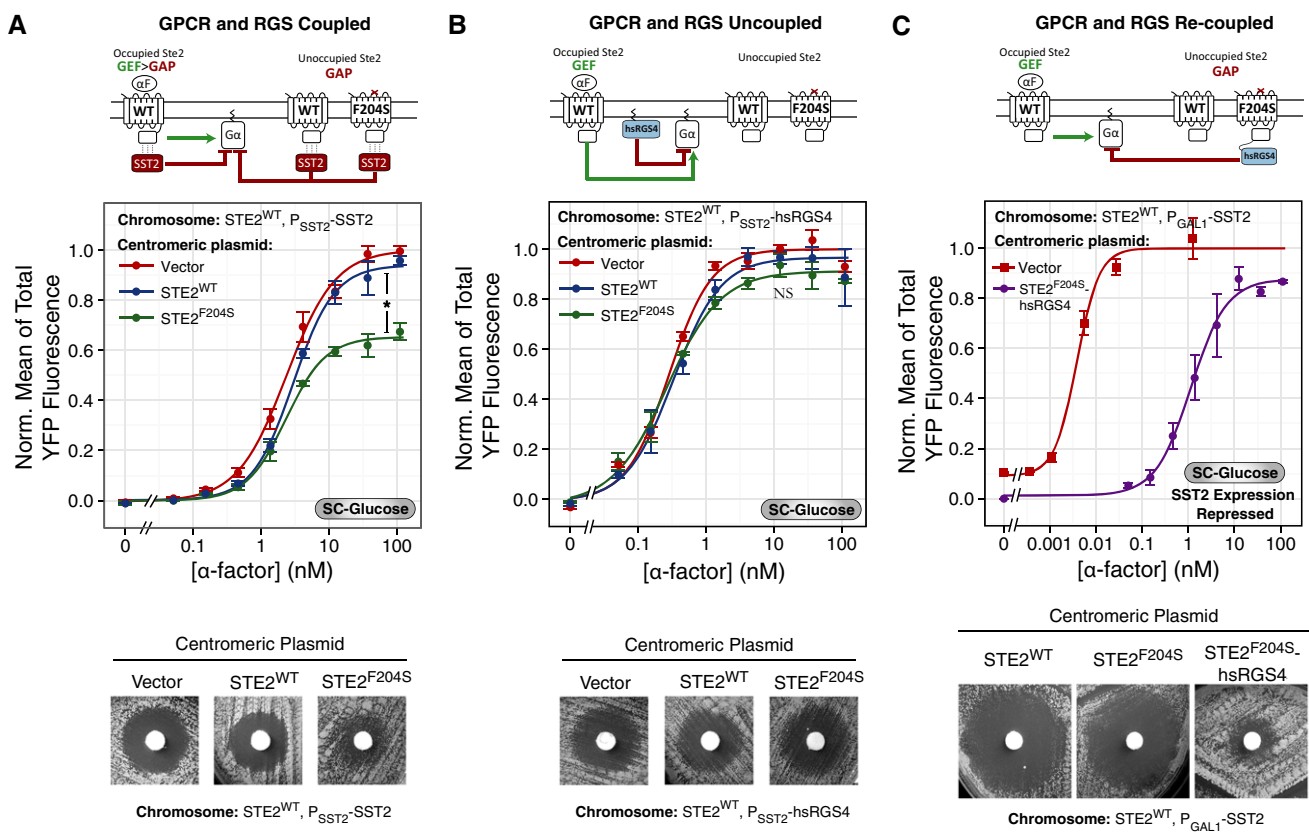

**Figure 7. Fractional occupancy measurement requires GPCR–RGS coupling.**

A–C   To alter the fraction of occupied receptors without changing the α-factor concentration, we co-expressed WT Ste2 with a mutant receptor unable to bind α-factor (STE2$^{F204S}$). We grew $P_{PRM1}$-YFP yeast with SST2 (A, YGV5666-68), hsRGS4-CFP (B, YGV5669-71), or $P_{GAL1}$-SST2 (C, ACY5662) in SC-glucose and transformed with an empty CEN-ARS plasmid (red), a plasmid with the STE2 gene (blue), the STE2$^{F204S}$ mutant (green), or STE2$^{F204S}$-hsRGS4-CFP (only in C, violet). We stimulated them with the indicated α-factor concentrations and 10 μM 1NM-PP1 for two hours and then imaged to measure accumulated YFP reporter. Data correspond to the mean ± SEM normalized YFP fluorescence of three independent experiments. We also performed halo assays with the same strains (bottom). In (C), we also include the halo of the strain that co-expresses STE2$^{WT}$ with STE2$^{F204S}$ in the absence of any RGS (ACL5682 and ACL5683). In this strain, STE2$^{F204S}$ is unable to reduce sensitivity.

Data information: For statistical significance, we compared the coefficients obtained by fitting data to a Hill-function model using non-linear mixed-effects models. In (A), the amplitudes of strains carrying a plasmid with STE2$^{WT}$ and STE2$^{F204S}$ were 0.92 ± 0.02 and 0.65 ± 0.03, respectively ($P < 10^{-4}$). In (B), STE2$^{WT}$, 0.97 ± 0.02; and STE2$^{F204S}$, 0.93 ± 0.03 ($P = 0.160$). In (C), in strains with SST2 expression repressed, we compared the EC$_{50}$s obtained with empty vector or a vector carrying STE2$^{F204S}$-hsRGS4: (3.8 ± 0.4) pM vs. (1.1 ± 0.3) nM, respectively.

mechanism of inhibition as well. We obtained similar results when assaying the sensitivity of the above strains to α-factor-induced cell cycle arrest in halo assays (Fig 7A–C, lower panels). Co-expression of the DN Ste2-F204S receptor reduced the sensitivity (smaller non-growth region) of strains that expressed Sst2$^{RGS}$ (WT), but not that expressed hsRGS4-CFP. Similarly, yeast expressing the Ste2-F204S-hsRGS4-CFP chimera formed small halos.

Taken together, these results strongly argue that a main mechanism why non-binder receptors are dominant-negative is that they recruit the RGS to the membrane where it can interact with G proteins.

### Global analysis of the *carousel* model identifies key constraints for *fraction measurement*

So far, our model analysis was based on the reference parameters (Table 1), and a limited analysis of the effect of changing two

parameters, receptor–Gα dissociation rate and localization of the RGS activity to the receptor. That analysis already showed relations between the value of these particular parameters and others that significantly altered the model behavior. Thus, to reveal the behaviors the model could exhibit and the relations between the parameter values that enable them, we explored a broad region of the parameter space. We varied each parameter logarithmically, scanning 8 orders of magnitude centered on the reference parameter.

Thus, we determined for which sampled points in parameter space the model shows robustness to changes in receptor abundance, that is, if it reports the fraction of occupied receptors. To do this, we simulated steady-state free-Gβγ DoR curves for different receptor abundances, for each parameter point sampled. Robustness to receptor abundance requires that the *amplitude* (the difference in free Gβγ between maximum and zero ligand) and the EC$_{50}$ of the DoR be insensitive to changes in receptor abundance. Thus, we

classified the observed behaviors into those that show *robust amplitude*, *robust $EC_{50}$*, and *robust* response (those that show simultaneously both partial robustness behaviors), which is the main focus of the simulation effort (see Section 3.6 of the Appendix for details). We then analyzed the parameters and their relations that gave rise to each of the three categories.

Of the $10^5$ parameter points sampled, 7.2% showed *robust $EC_{50}$* while only 1.8% showed *robust amplitude*, suggesting that the latter condition is harder to achieve than the former. Of the sampled points, 1.1% showed a *robust response* (both *robust amplitude* and *robust $EC_{50}$*) and therefore responded to the fraction of occupied receptors.

In order to visualize the distribution in parameter space of these 1.1% of sampled points, we used a matrix of 2D histograms (Fig 8A). Each panel in the matrix corresponds to the projection on the plane defined by two parameters of this subset that shows an overall response robust to variations in receptor abundance. As shown in Fig 8A, some 2D histograms show a homogeneous distribution of points (*e.g.,* panel R04: $\log_{10}\left(k_{Ef}^{RG}\right)$ vs. $\log_{10}(k_{off}^{L\cdot R})$), while others show a conspicuous structure with regions devoid of points (*e.g.,* panel R16: $\log_{10}\left(k_{Hf}^{Gt}\right)$ vs. $\log_{10}(k_{off}^{R\cdot G})$) . For each panel in which a clear structure can be identified, we can define a *restriction* between two parameters that has to be satisfied for the model to respond to the fraction of occupied receptors. For example, based on panel R16 of Fig 8A, the restriction $k_{Hf}^{Gt} \ll k_{off}^{R\cdot G}$ has to be fulfilled (see details in the Appendix). Restated, this restriction means that the probability that a free G$\alpha$ subunit (not coupled to a receptor) converts its bound GTP to GDP during the average receptor–G$\alpha$ interaction time has to be very small for the model to show robustness to receptor abundance (Fig 8C).

We found several significant restrictions between parameters, satisfied by at least 95% of the points, evident in the set that shows a response dependent on the fraction of occupied receptors (shown by diagonal lines in Fig 8A and Appendix Fig S6, and Appendix Table S7). Some of these are *necessary* restrictions, meaning that they are satisfied by 100% of the points in parameter space that show a response robust to changes in the number of total receptors (R13, R45, R67, and R89; solid lines in Fig 8A). Of these restrictions, most are already required for the model to show either of the partial behaviors of *robust amplitude* or *robust $EC_{50}$* (gray lines in Fig 8A; see also Section 3.6 of the Appendix). However, we found that one *necessary* restriction, R67, was required for robustness to receptor abundance and not needed for either of the partial behaviors of *robust amplitude* and *robust $EC_{50}$*, namely that $k_{Hf}^{Gt} \ll k_{Hf}^{LRGt}$ . This restriction indicates that the GTP hydrolysis rate of G$\alpha$ has to

be greatly increased when G$\alpha$ is coupled to a receptor (Fig 8D). Interestingly, this is exactly what we expect if the RGS is active only when physically associated with the receptor. We therefore call this condition the *localized RGS* restriction. This restriction, together with R35 ($k_{Ef}^{G} \ll k_{Ef}^{LRG}$, that G$\alpha$ exchanges GDP with GTP much faster when coupled to a ligand-bound receptor than when uncoupled) in Fig 8A, makes it unlikely for a G$\alpha$ to change its state while uncoupled from a receptor.

Only 12% of the points that show a response robust to the number of receptors also show DoRA. These points satisfy a new *necessary* restriction (R07: $k_{off}^{L\cdot R} < k_{Hf}^{LRGt}$; Appendix Fig S7 and Fig 8E), involving the off-rate of the receptor–ligand binding reaction, the only parameter that had no restrictions until now. It says that the receptor has to remain occupied by the ligand enough time to allow for the hydrolysis of GTP by a G$\alpha$ that is bound to it to take place.

Remarkably, upon closer examination of the relationship between the ligand–receptor off-rate and the other parameters for DoRA, we found that points for which the ligand–receptor off-rate is faster than the uncoupling of G$\alpha$ from the receptor ($k_{off}^{L\cdot R} > k_{off}^{R\cdot G}$) fall close to the identity in panel R07 (i.e., $k_{off}^{L\cdot R} \approx k_{Hf}^{LRGt}$ ; green circles in Fig 8B) and therefore have ligand–receptor off-rates only slightly smaller than the GTP hydrolysis rates of receptor-coupled G$\alpha$. On the other hand, points with uncoupling of G protein from receptor faster than the ligand off-rate ($k_{off}^{R\cdot G} > k_{off}^{L\cdot R}$) fall over the identity in panel R17 (i.e., $k_{off}^{R\cdot G} \approx k_{Hf}^{LRGt}$ ; violet triangles in Fig 8B), so the GTP hydrolysis rate of receptor-coupled G$\alpha$ is approximately equal to the receptor–G$\alpha$ uncoupling rate.

In summary, for combinations of parameters that show robust DoRA, the GTP hydrolysis rate of receptor-coupled G$\alpha$ has to be similar to the largest of the ligand and G$\alpha$ dissociation rates from the receptor ($k_{Hf}^{LRGt} \approx \max\left(k_{off}^{L\cdot R}, k_{off}^{R\cdot G}\right)$). In biological terms, this condition requires that on average, only one GTP hydrolysis event occurs during the lifetime of a ligand–receptor–G$\alpha$ ternary complex.

## Discussion

This work was motivated by the observation that if the response of the pheromone pathway is robust to changes in receptor abundance (Blumer *et al*, 1988; Konopka *et al*, 1988; Reneke *et al*, 1988; Konopka & Jenness, 1991; Shah & Marsh, 1996; Leavitt *et al*, 1999; Gehret *et al*, 2012) (Fig 2), then this pathway has an output that depends on the fraction of occupied receptors. To be able to respond to the fraction of occupied receptors, the system needs to measure

---

**Figure 8.  Global analysis of the *carousel* model reveals parameter restrictions for a robust response.**

A    Matrix of 2D histograms for points in parameter space that show an overall response robust to receptor abundance. Each panel shows the distribution of the points projected on the plane defined by the parameters indicated in the top and left margins, for the x- and y-axes, respectively. Scales indicate the $\log_{10}$ of each parameter. Bins span one log in each direction and are colored according to the frequency color guide. In the diagonal of the matrix, the 1D histograms of the $\log_{10}$ of the parameters are shown. Lines representing restrictions between parameters are shown in some panels. Solid lines represent *necessary* restrictions. Gray lines indicate restrictions required for having a normal amplitude, robust $EC_{50}$, or robust amplitude of the DoR curve (see Appendix Supplementary Methods). Red lines are novel restrictions required for a robust response. Each panel is labeled with an R followed by two digits depending on the position of the panel, for easy reference.

B    2D histograms involving parameters $k_{off}^{L\cdot R}$, $k_{off}^{R\cdot G}$, and $k_{Hf}^{LRGt}$ for points that show a robust response and DoRA. In each panel, a black dashed line shows the identity. Points above of the identity in panel R01 ($k_{off}^{R\cdot G} > k_{off}^{L\cdot R}$) are plotted as violet triangles, while points below this line ($k_{off}^{R\cdot G} < k_{off}^{L\cdot R}$) are plotted as green circles in the three panels.

C–E  Schematic representation of restriction R16. Reactions involving the pertinent rates are illustrated and highlighted in the *carousel* scheme. (D) Same as in (C) for restriction R67 (localized RGS). (E) Same as in (C) for restriction R07, required for robust DoRA.

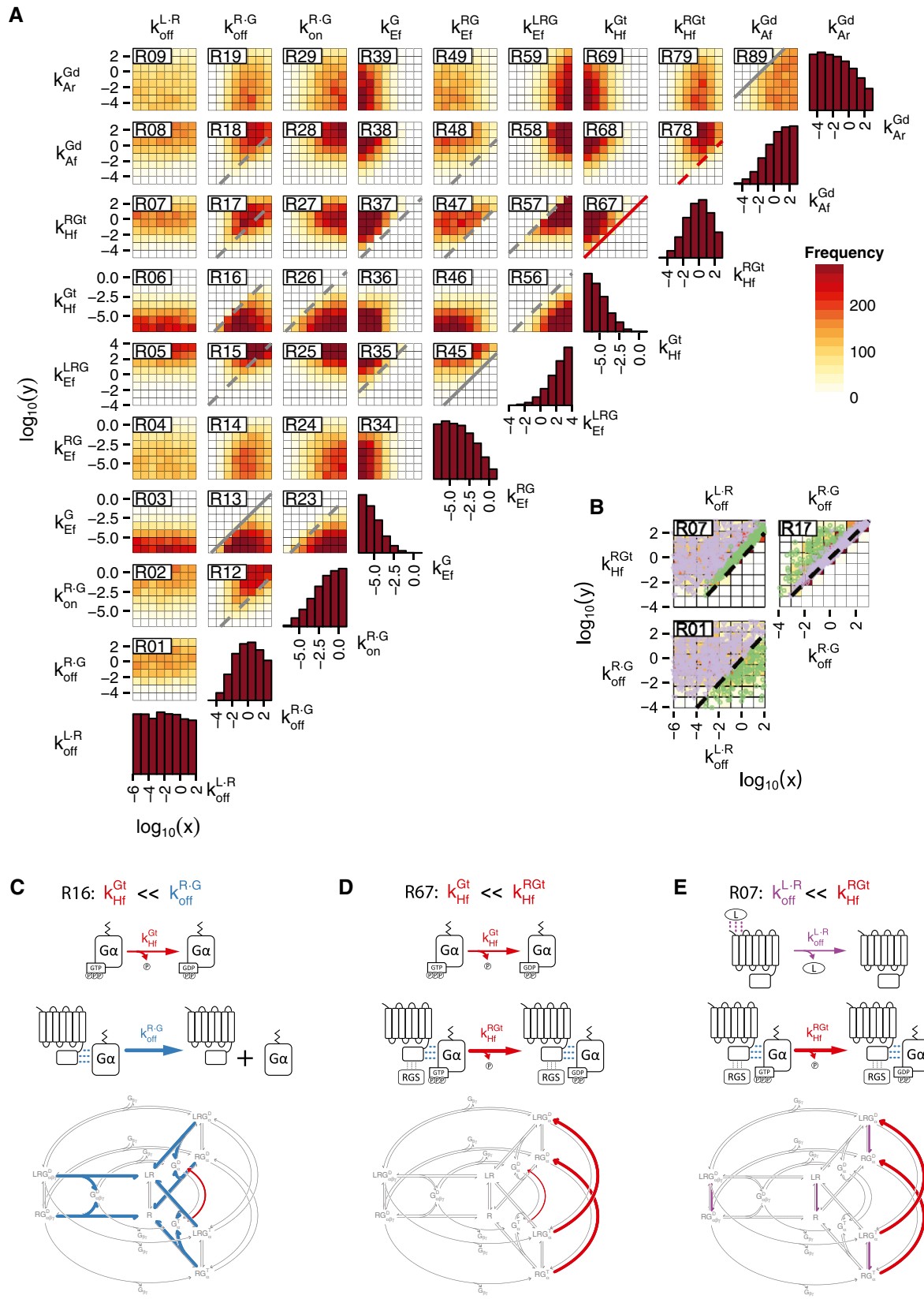

**Figure 8.**

the number of both occupied and unoccupied receptors. In addition, the PRS responds linearly to this fraction, resulting in DoRA (Yu *et al*, 2008). Both these system-level properties (fraction measurement and DoRA) are interesting, as they allow a precise transmission of extracellular agonist concentration in the face of variations in receptor abundance.

We mapped experimentally the point of action of the fraction measuring mechanism upstream of Ste5's membrane recruitment, to the interaction between the receptor and the G protein. Thus, to explore how this system computes the fraction of occupied receptors, we developed the thermodynamically complete *carousel* model of G-protein activation. This model extends the TCM to incorporate the different possible states of the G protein. The *carousel* model captures the *precoupled* (or *ternary complex*) and *collision-coupling* (or *catalytic reaction*) regimes proposed in the literature for GPCRs (Lauffenburger & Linderman, 1993; Roberts & Waelbroeck, 2004), and it can also represent the physical interaction between receptors and RGSs (Ballon *et al*, 2006; Neitzel & Hepler, 2006).

Analysis of the behavior of this model led to the core prediction that the activity of the RGS has to be localized to the receptor for the system to be able to measure fraction of occupied receptors (Fig 8). In such a view, a receptor–RGS complex is a *paradoxical component* that catalyzes antagonistic reactions (Hart & Alon, 2013): Ligand-occupied receptors act as activators (*push*), while unoccupied receptors act as inhibitors (*pull*). Intuitively, the *push-pull* (Andrews *et al*, 2016) nature of the receptor–RGS complex then suggests a mechanism by which the G protein can respond to the fraction of occupied receptors. If uncoupled G$\alpha$ subunits are not likely to exchange or hydrolyze their bound guanine nucleotide, then their activation state will be determined by the occupancy state of the last receptor–RGS complex they interact with. In this model, if we are in the *collision-coupling* regime, G$\alpha$ subunits randomly interact with ligand-occupied and unoccupied receptor–RGS complexes; consequently, the fraction of active G$\alpha$ will depend on the fraction of occupied receptors (Fig 3F). Thus, the Ste2$^{GPCR}$–Sst2$^{RGS}$ complex operates as a ratiometric sensor, and thus by definition is robust to changes in its abundance. Note that due to the way the RGS is encoded (as a rate of GTP hydrolysis by G$\alpha$), in the model there is always enough RGS for any receptor abundances simulated. However, experimentally, this might not be the case. In the PRS, before stimulation there is a similar number of Sst2$^{RGS}$ and Ste2$^{GPCR}$ molecules (Ghaemmaghami *et al*, 2003). Thus, when Ste2$^{GPCR}$ is overexpressed, it is possible that there is not enough Sst2$^{RGS}$ to form complexes with all receptors, potentially preventing the push-pull mechanism to operate. However, the interaction between Sst2$^{RGS}$ and Ste2$^{GPCR}$ does not have to be stable for the system to work. If complexing is fast enough, one Sst2$^{RGS}$ might visit and act as GAP on more than one Ste2$^{GPCR}$. There is no published binding rate for this interaction, but the binding does not seem to be very tight, since Sst2-GFP fusions show a mainly cytoplasmic staining [see, e.g., Ballon *et al* (2006)].

To test the predictions of our model, instead of eliminating the RGS (or using an Sst2 mutant with reduced affinity for the receptor, such as *sst2-Q304N* (Ballon *et al*, 2006), which would have resulted in a cytoplasmically localized Sst2 unable to act on Gpa1, functionally equivalent to a $\Delta sst2$; Ballon *et al*, 2006), we sought to delocalize it within the plane of the membrane, such that it would act homogeneously on all G$\alpha$ subunits independently if they are

coupled or not to receptors. To this end, we replaced endogenous SST2$^{RGS}$ ORF with the ortholog hsRGS4, which is a GAP for Gpa1$^{G\alpha}$ and localizes to the plasma membrane in a manner independent of receptors. This approach had the added advantage over the deletion of SST2$^{RGS}$ that it resulted in strains with similar sensitivity to pheromone as WT. In our experiments, we use $\Delta bar1$ cells, which gives us a good control over the external pheromone concentration. But the extra deletion of SST2$^{RGS}$ renders cells extremely (~1,000-fold) sensitive to pheromone, greatly complicating experiments in $\Delta sst2\Delta bar1$.

Our most direct test of the model's prediction was the measurement of Ste5 recruitment, since that event directly follows G-protein dissociation and may be measured in the first minutes after stimulation, avoiding the complications originating from feedback regulation. In these direct tests, as predicted by the *carousel* model, strains with hsRGS4 had a response that increased with receptor abundance and therefore were unable to measure fraction of occupied receptors (Fig 4A–C). Following our model, we then succeeded at restoring robustness in cells expressing hsRGS4 by recruiting it to Ste2$^{GPCR}$ (Fig 4D–F). We obtained similar but more complex support for the model's *push-pull* hypothesis using the longer-term transcriptional reporters (Figs 5–7). In this case, we restored robustness by directly fusing Ste2$^{GPCR}$ to hsRGS4, bypassing the need of the DEP domain of Sst2$^{RGS}$, suggesting that the endocytosis protective function attributed to Sst2$^{RGS}$ (Venkatapurapu *et al*, 2015) that resides in this domain was not required for robustness to changes in receptor number.

Several published results support an inhibitory role of the unoccupied receptor–RGS complex, suggested by the *carousel* model: (i) In the original screen for mutants that do not arrest the cell cycle in response to $\alpha$-factor, Hartwell found that *ste2*$^{GPCR}$ mutants elevate a-factor production in MATa cells by 250%, while all other *ste* mutants reduced this secretion (Hartwell, 1980); (ii) basal signaling is increased in $\Delta ste2$$^{GPCR}$ cells (Sommers *et al*, 2000), a phenotype complemented by episomal expression of STE2$^{GPCR}$, but not of C-terminally truncated *ste2-T326* (Dosil *et al*, 2000); (iii) C-terminal truncation of the receptor results in higher basal signaling and adaptations defects, and these phenotypes are recessive to WT receptor (Konopka *et al*, 1988; Reneke *et al*, 1988); (iv) basal signaling by constitutively active receptor alleles diminishes if co-expressed with WT receptors (Dosil *et al*, 2000; Sommers *et al*, 2000; Gehret *et al*, 2012), but increases if co-expressed with C-terminally truncated receptors (Gehret *et al*, 2012); (v) the observation that mutant alleles that do not bind pheromone exert dominant-negative (DN) effects (Dosil *et al*, 1998, 2000; Leavitt *et al*, 1999; Gehret *et al*, 2012); and (vi) no DN effects are observed if the expressed non-binding receptor alleles are C-terminally truncated (Dosil *et al*, 2000; Gehret *et al*, 2012), or are expressed in $\Delta sst2$$^{RGS}$ cells (Gehret *et al*, 2012).

So far, there was no clear mechanism by which receptor alleles with low/no affinity for $\alpha$-factor inhibit, nor of the role played by Sst2$^{RGS}$ in this inhibition. Dosil *et al* (1998) originally suggested that DN receptors acted by sequestering G proteins away from WT receptors, since they were able to rescue the DN effect by overexpressing the three subunits of the G protein. However, recently, Gehret *et al* (2012) put that hypothesis into question by showing that DN receptors are still able to inhibit Ste2$^{GPCR}$-Gpa1$^{G\alpha}$ chimeras. Instead, they postulated that DN receptors act when forming heterodimers with WT receptors by some conformational change. Here, in another test

of our model, we showed that what is needed for DN activity is the interaction of an RGS with the receptor to create an inhibitory complex (Fig 7).

Our modeling analysis indicated that the push-pull topology created by the Ste2$^{GPCR}$–Sst2$^{RGS}$ complex was essential for fraction measurement in the low range of Ste2$^{GPCR}$ abundances but not in the high range. Our detailed experimental exploration verified this prediction. Two results were not predicted by the *carousel* model. The first one was the mild inhibition observed when Ste2$^{GPCR}$ was overexpressed. This inhibition was independent of RGS localization (Fig 5), and it was detectable even in the absence of any RGS (Fig EV4D). Some (but not all) previous works also show some degree of inhibition upon overexpression of Ste2$^{GPCR}$ (Konopka & Jenness, 1991; Leavitt *et al*, 1999). This inhibition could be explained if G-protein activation required a third component besides the bound receptor and the G protein itself. Then the likelihood of occurrence of such a ternary complex would diminish if Ste2$^{GPCR}$ were in excess. Alternatively, inhibition might be due to spatial effects: Excess Ste2$^{GPCR}$ might not be able to localize correctly at the signaling/polarity site, and thus, mislocalized Ste2$^{GPCR}$ might titer out G proteins.

The second unpredicted result was that the robustness of the PRS transcriptional reporter to changes in Ste2$^{GPCR}$ abundance in the low receptor abundance range, which required the RGS to be complexed with the GPCR, was only evident at concentrations of α-factor equal to or lower than the $K_d$ between the ligand and Ste2$^{GPCR}$. What happens at higher concentrations? One possibility is that in normal cells at high concentrations of α-factor, the Ste2$^{GPCR}$–Sst2$^{RGS}$ interaction might be weakened due to hyperphosphorylation of the Ste2$^{GPCR}$ C-terminal tail, which might reduce its affinity for Sst2$^{RGS}$ (Ballon *et al*, 2006). Another non-exclusive possibility stems from the fact that in these experiments, the membrane abundance of Ste2$^{GPCR}$ at high α-factor concentrations is lower than at low doses (Fig EV1F). This differential abundance is the result of combining α-factor-independent expression out of the $P_{GAL1}$-STE2$^{GPCR}$ gene (dependent on the concentration of β-estradiol only; Fig EV1F and G) with α-factor-modulated endocytosis (Fig EV1F) (Jenness & Spatrick, 1986). As a consequence of this, at high α-factor we in fact tested a range of lower Ste2$^{GPCR}$ abundances than at low α-factor. It is possible that at that low abundance range, robustness collapses, even with a working Ste2$^{GPCR}$–Sst2$^{RGS}$ complex. Supporting the first possibility (α-factor modulated Ste2–Sst2 interaction), we were able to rescue the lost robustness at high α-factor using the Ste2$^{GPCR}$-RGS4$^{RGS}$ chimera, in which the RGS is permanently attached to the GPCR. A surprising experimental result obtained using this fusion strain was remarkably captured by the *carousel* model: At low concentrations of α-factor, increasing Ste2$^{GPCR}$ abundance inhibited signaling. In the model, where the RGS activity is associated with the GPCR, this is because at low receptor abundance there is not enough RGS activity to counteract the spontaneous activation of G protein. This same reason might explain the experimental results, since inhibition of transcription by increasing Ste2$^{GPCR}$-RGS4 synthesis is observed even in the absence of α-factor, suggesting that at the lowest β-estradiol concentrations there is not enough Ste2$^{GPCR}$-RGS4 chimera to inhibit all the spontaneously activated G proteins.

Interestingly, we have recently shown that push-pull topologies such as that of the Ste2$^{GPCR}$–Sst2$^{RGS}$ complex can result in perfect DoRA in signaling pathways (Andrews *et al*, 2016). According to

the *carousel* model, besides push-pull, in order for there to be DoRA, the hydrolysis rate of receptor-coupled Gα has to be within a narrow range, determined by the maximum between the ligand–receptor and Gα–receptor off-rates (Fig 8B). Because of the very slow unbinding of α-factor from its receptor (Jenness *et al*, 1983; Bajaj *et al*, 2004), in the PRS this means that the dissociation rate of receptor from Gα and the hydrolysis rate of GTP by Gα coupled to receptor have to be roughly similar. This seemingly restrictive condition could be ensured if these two reactions were mechanistically coupled at the molecular level.

Cells use several mechanisms to attenuate the effects of intrinsic and extrinsic variability (Thattai & van Oudenaarden, 2001; Swain, 2004). Measuring the fraction of occupied receptors is another such mechanisms (Fig 6D), avoiding the propagation of cell-to-cell variability in receptor abundance to the pathway's output. Previous results indicated that Sst2$^{RGS}$ has a noise-suppressing function (Siekhaus & Drubin, 2003; Dixit *et al*, 2014).

We think that the ratiometric mechanism we observed is widespread, since physical interactions between RGS and GPCRs are fairly common in these kinds of signaling systems (Neitzel & Hepler, 2006). Besides giving specificity to the RGS activity, this interaction allows the system to operate in a ratiometric mode and thus to be robust to changes in receptor abundance. It is possible that this property of GPCR signaling systems is what makes them adequate to accurately measure extracellular ligand concentrations, which could in turn help explain why they are so widespread in eukaryotes.

# Materials and Methods

### Strains and media

All strains used in this study were derived from ACL379 (Colman-Lerner *et al*, 2005) strain (W303-1a, MATa, Δ*bar1*) using standard methods (Fink & Guthrie, 1991) (see Appendix Table S1). Cells were grown to exponential phase in liquid synthetic complete media BSM (Q-Bio gene; MP Biomedicals) with either 2% glucose (SC-Glu), or 2% galactose and 1% raffinose (SC-Gal/Raff). Preparation of cells for cytometry is detailed in the Appendix.

### Modeling

The *carousel* model was implemented in COPASI (Hoops *et al*, 2006), exported as a C-function, and automatically compiled and executed from R (R Core Team, 2016). We did a *Latin hypercube sampling* (McKay *et al*, 1979) of parameter space, classified the results according to their behavior to changes in the abundance of total receptors, and did the restriction analysis described in the text (see Appendix Supplementary Methods for more details).

### Statistical methods

Experiments shown in the main figures were done in at least three biological replicates, except for the binding of fluorescent α-factor shown in Fig 2A due to the limited supply of this reagent. Error bars correspond to the 95% confidence interval of the mean (CI95) or to the standard error of the mean (SEM), as indicated. For statistical

significance determination, we used ANOVA, followed by a Tukey post-test when appropriate, or the non-linear mixed-effects analysis in the cases where we fit dose-response data to a Hill-function model.

**Expanded View** for this article is available online.

## Acknowledgements
We thank P. Pryciak, R. Brent, A. Ventura, V. Repetto, A.V. Grande, W. Peria, and S. Andrews for discussion and/or comments on the manuscript; D. Drubin for kindly providing fluorescent α-factor; G. Pesce for providing the GEV plasmid; and P. Pryciak for providing a strain with Ste5-YFPx3 and the DN Ste2 plasmid. This work was supported by grants PICT2010-2248 and PICT2013-2210 from the Argentine Agency of Research and Technology, and grant 1R01GM097479-01, subaward 0000713502, from the National Institute of General Medical Sciences, National Institutes of Health.

## Author contributions
AB, GV, and AC-L designed research; AB, GV, PD, AC, MB, and ILP performed research; and AB, GV, and AC-L analyzed the data and wrote the manuscript.

## Conflict of Interest
The authors declare that they have no conflict of interest.

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
