## [Review Process File · Molecular Systems Biology]

Yeast GPCR signaling reflects the fraction of occupied receptors, not the number

Alan Bush, Gustavo Vasen, Andreas Constantinou, Paula Dunayevich, Inés Lucía Patop, Matías Blaustein and Alejandro Colman-Lerner

Corresponding author: Alejandro Colman-Lerner, IFIBYNE-CONICET/University of Buenos Aires

Review timeline:

Submission date:	25 February 2016
Editorial Decision:	14 April 2016
Additional Correspondence:	14 May 2016
Revision received:	31 October 2016
Accepted:	22 November 2016

Editor: Thomas Lemberger

Transaction Report:

1st Editorial Decision

14 April 2016

Thank you again for submitting your work to Molecular Systems Biology. We have now heard back from three of the four referees who agreed to evaluate your manuscript. Rather than delaying further the process, I prefer to make a decision now with the available reports, given that the overall recommendations are similar. As you will see from the reports below, the referees find the topic of your study of potential interest. They raise, however, several important concerns on your work, which should be convincingly addressed in a major revision of the study.

The recommendations provided by the reviewers are very clear. One of the recurrent points raised by the three reviewers refers to issues related to the use of the human RGS4. In the light of the reviewers's comments it would therefore be important to perform some of the key experiments with a mutant of Sst2 that cannot interact with Ste2 (reviewer #2 provides a concrete example of such a mutant).

REFeree REPORTS

Reviewer #1:

This is a quite interesting work that presents a nice model that can explain how a G-protein response can track the fraction of GPCRs that are bound by ligand, as opposed to the total number of bound receptors. In so doing, the authors are able explain many published experimental observations in the yeast system. The paper also contains very high quality quantitative data, as it typical of work from Dr. Colman-Lerner's lab.

Major Points

1. From the point of view of modeling, there are some points that have been glossed over. The model nicely explains how the amount of active Galpha can be made responsive to the fraction of occupied receptors. However, in the yeast system it is beta-gamma that transmit the signal to Ste5 and Ste20, so several other parameters must of necessity play a role, particularly the rate of recapture of beta-gamma by alpha-GDP, and the extent to which this is inhibited by the binding of beta-gamma to Ste5 and Ste20.

2. On the experimental side, why didn't the authors analyze an Sst2 mutant that can no longer bind to Ste2 due to DEP-domain mutations? Also, did the authors consider attaching Sst2's DEP domain to the hsRGS4?

3. How long were the cells exposed to pheromone before the readings were taken? Was it long enough that the pheromone inducibility of the wild-type Ste2 receptor could have diminished the difference between the WT vs GAL promoters?

If the above points can be adequately addressed then I would be willing to change my rating for "Validity of Conclusions Drawn" from Medium to High.

Minor Points

1. It is not completely clear that the long review of receptor theory in the introduction is completely necessary.

2. Introduction: "and that it is this agonist-receptor complex produces the measured effect."

Change to

"and that it is this agonist-receptor complex that produces the measured effect." Or

"and that this agonist-receptor complex produces the measured effect."

3. page 4: "which allowed to explain the different apparent affinities observed for some ligands"

Change to

"which allowed them to explain the different apparent affinities observed for some ligands" Or

"explains the different apparent affinities observed for some ligands" Or

4. page 12: "independent on the state of G"

Change to "independent of the state of G"

Reviewer #2:

Overall, The paper is interesting and provides new insights into how Ste2 (and maybe other GPCRs) regulate signaling through the G protein trimer. Here are my further thoughts and comments:

The research appears to have been well designed, executed, and clearly communicated for the most part. The paper helps to explain more precisely how pheromone signaling appears to be insensitive to receptor abundance, and attributes this to Sst2's direct interaction with Ste2 by using a simplified mass-action model and an experimental verification.

The model itself is similar to previous G protein activation models (the authors reference a couple of them), but proposes to account for Sst2's localized GAP activity by maximizing the rate of Galpha-GTP hydrolysis when Galpha is associated with the receptor, as opposed to when it is not associated to the receptor. The rate assignments and simplifying assumptions in the model appear to be sound. One of their key assumptions is that the receptor can bind to Galpha, regardless of whether it is bound to Gbeta. This has not been experimentally validated, but may be reasonable given that they define the GTP hydrolysis rate to be different based on whether Galpha is bound to receptor or not. Specifically, they define the hydrolysis rate when bound to the ligand-occupied receptor to be the same when bound to the unoccupied receptor, but both of these rates are higher than when the G protein is not bound to the receptor at all. They also do a parameter analysis to show that this particular rate asymmetry is needed to satisfy both robustness to receptor abundance and DoRA.

The use of hsRGS4 as an experimental validation for the model is well motivated in the text, but there are a few clarifications that are needed:

1) The result in Fig 5a makes sense, but in Fig 5b, it is not clear why higher receptor expression results in reduced signaling for the SST2 + hsRGS4 strain. It seems their intention is to show that signaling does not increase, but the fact that there is a substantial decrease in signaling, probably warrants some type of explanation.

2) Instead of hsRGS4, why not use native Sst2 with the Ste2 docking domain disrupted? The allele, *sst2(Q304N)*, accomplishes this (Ballon et al, 2006) and has been used in other reports.

3) Since only two receptor levels are shown (in Fig 5b and 5c), it is not clear what the relationship between receptor level and signaling is like in the hsRGS4 strain. Is it linear or graded? Does WT (grown in glucose) fall in between the two signaling levels given?

The analysis of cell-to-cell variability in the hsRGS4 strains is interesting and consistent with what would be expected from recent reports (particularly from Dohlman's lab), but the description of the analysis is a little unclear.

1) When reporting the variation statistic values, they do not specify at what dose the gene expression variation represents (for the SST2 strain). Since it does not change much with dose, was the value at 0nM used and then compared to the basal receptor variation values?

2) It would be informative to compare the signaling variation of the *sst2del* hsRGS4 strain with the regular *sst2del* strain, to see if variation is actually maximized by only removing the Sst2-Ste2 interaction. Some might expect variation to increase further if Sst2 is fully removed.

Reviewer #3:

This manuscript describes both experimental and systems modeling approaches for understanding important properties of the yeast pheromone response pathway that are likely to have general implications for G protein coupled receptor-dependent signaling pathways. The paper focuses, in particular, on modeling the lack of variation in signaling responses as the number of pheromone receptors expressed at the cell surface is varied and on dose-response alignment, linear correlation of signaling at different stages of the pathway with the occupancy of receptors by agonist. A strong aspect of the manuscript is that the modeling leads to predictions about the system, particularly the importance of localization of the RGS protein Sst2p to receptors, that are then experimentally tested by assaying for the effects on signaling of introducing a mammalian RGS protein that is not expected to be localized to receptors. The results are important in view of the fact that the pheromone pathway has been used as a test bed for evaluating procedures for quantitative analysis of signaling pathways. The experimental parts of the paper appear to be generally well-performed and validated using different approaches. The considerable strengths of the paper provide a strong basis for publication, however, before this can happen it seems to me that several significant issues should be addressed by the authors:

Major points:

1) A major concern is that, although the action of Sst2p plays a critical role in the model, the abundance and properties (such as on- and off-rate for receptor and/or G protein, acceleration of hydrolysis rate, etc.) of this protein do not appear to be explicitly considered in the modeling beyond a generalized effect on GTP hydrolysis. The paper concludes that stable association of Sst2p with receptors is required for the observed ratiometric signaling over a range of receptor concentrations, however, published estimates of Sst2p abundance place it at or below the abundance of even chromosomally-encoded receptors under the normal STE2 promoter. It is difficult to understand why this apparently critical component is not more explicitly considered in the modeling.

2) Some serious concerns arise in considering the critical experiment involving expression of human RGS4:

a) To what extent are observed effects in the *sst2* deletion strain due to the deletion vs. to the expression of RGS4, since no control without RGS4 is shown? RGS4 seems to be doing something to affect signaling, since its expression slightly decreases basal signaling in cells containing Sst2p in Fig. 4a, but its effect on overall dose response is not clear. Also, maximal signaling levels in Fig. 5b and Fig. 5c are decreased compared to Fig. 1b and 1c (is this significant?) Deletion of SST2 would be expected to decrease the EC50, as observed for the PGAL1-STE2 strain in Fig. 5c, simply by removing the RGS. How do we know whether this is due to loss of localization or simply to partial loss of RGS activity? The fact that increased sensitivity is not seen with the WT strain is interesting, but doesn't directly answer this question.

b) A related concern is that, contrary to what is stated in the paper, much of the RGS4 does not seem

to be localized to the membrane, and thus, may not be available even in delocalized form, based on Fig. S2b.

3) There is uncertainty about interpreting the experiments determining the number of receptors on the cell surface;

a) Despite the addition of metabolic inhibitors, there is continued endocytosis, as is evident from Fig. 1a (bottom). Why were cells incubated with ligand for 3-5 hours at 30 degrees, which certainly exacerbates this problem? Material irreversibly trapped by endocytosis should not be included as cell surface expression of receptors available for binding.

b) The described quantitation of binding is confusing. The main text refers to a "reference value" for WT Ste2p in glucose which apparently applies to the strains in Fig. 2, which, for some reason, contain fluorescent protein insertions at the actin locus. Why? Similar comparisons for strains in Fig. S1a (for example ACL394 vs. YAB3502) give very different numbers.

c) An additional complication in the quantitation of sites is the fact that most strains used to monitor binding of HiLyte488-tagged ligand also contain YFP fused to PRM1 or STE5 and/or CFP fused to ACT1 or hsRGS4. The contributions of these proteins to binding measurements made using the YFP cube are not discussed or, apparently controlled for.

4) The 5-fold difference between the high and low Ste2p-expressing strains used in the manuscript is rather small compared to differences obtained by other means. There is also concern that many aspects of cell metabolism, including some possibly affecting signaling, are known to change in comparing glucose vs. raf/gal-grown strains. While measurement of levels of various components under the different conditions described in the paper is a useful first step, there are other important avenues by which the different conditions could affect signaling, such as changes in phosphorylation states of components.

5) Glucose-repressed PGAL1-STE2 shows no response to alpha-factor (strains YAB5302 and YAB5313 in Figure S1c), contrary to previous results discussed in the text.

6) In comparing the precoupling vs. collision coupling models, one reason for disfavoring the precoupling model is that reducing the number of receptors below that of G proteins should lead to a decrease in signaling. However, decreased signaling by glucose-repressed PGAL1-STE2 is reported in this manuscript and in previous studies.

7) This manuscript essentially consists of two separate papers, one of them being the main text and the other being the supplementary material, which includes many important details and figures, with some duplication of points from the main text. This makes it hard to read. I'd be for including more material in the main text.

8) It would be helpful if the manuscript more explicitly compared the modeling approaches and results with the extensive previous modeling of this system.

Minor points:

1) The labeling for Figure 3b is deficient. What is the $K(RG)_{off}$ for panel b and how is it different from panel d?

2) Graphs showing dose response curves on a log scale that include "0" points should contain breaks in the axis.

3) The manuscript a number of grammatical errors and errors in the citations that should be corrected.

Additional Correspondence

14 May 2016

We have received an unexpected fourth delayed review of your manuscript. This report does not alter our decision. I am forwarding it to you in the hope that you will find it helpful.

Reviewer #4

In this article the authors address the observation that yeast pheromone signaling is robust to variation in the abundance of receptor. Since this is inconsistent with the simplest idea of a receptor-pheromone complex leading to signaling they propose a different mechanistic model for the G-protein activation and signaling cascade. They use theory, numerical simulation and experiments to support a model in which the regulator of G protein signaling (RGS) interacts with a and is activated by the pheromone receptor. This activation of the RGS's GTPase-activating function leads to the "activated" G-protein concentration being controlled by the ratio of receptor-pheromone complexes to unbound receptors. This ratiometric measurement makes the pheromone response robust to fluctuations in receptor abundance and rise linearly with the fraction of pheromone-bound receptors..

The work is clear in its approach and chooses to develop and test a specific mechanistic hypothesis using numerical modeling and experiments. The model is built from established molecular understanding of the pheromone signaling and G-protein signaling. They propose a coupled model with reasoned assumptions to reduce the parameter count. In particular they used micro-state reversibility and the independence of binding events to reduce the model to parameters that have been evaluated in previous literature.

In summary, this work builds a thermodynamically complete molecular model of the pheromone response system. This model is consistent with the observed robustness of pheromone signaling and they localize this robustness to the RGS-receptor interaction by numerical modeling and experimental testing. This work is interesting and worth publishing with minor changes to the writing. Specifically, the writing needs more clarity in how the hsRGS4 & SST2 expressing cells are explained based on the model. Additionally the reasoning for the experiments performed needs to be more complete (see below), potentially taking better advantage of literature as presented in the discussion

Comments

In the results section when discussing the initial experiments of supplementing Sst2 with over-expressed hsRGS4 it is necessary to comment further on why the amplitude, sensitivity, and noise suppression is still the same as WT. The current writing of the article suggests that the ratiometric measurement is contingent on making RGS activity depend on its interaction with the receptor (page 17, second paragraph). The authors need to be clearer that in this experiment, the majority of the GAP stimulation must come from receptor bound Sst2 rather than free hsRGS4 and do more to justify this argument.

The interpretation of their model and experiments is contingent on the fact that there is a sufficient pool of cytoplasmic Sst2 for all Ste2 to be in complex, while the free Sst2 has low GAP activity (to ensure that only receptor-bound GAPs are active). It would be prudent to discuss the appropriate literature (if available) or discuss the reasoning for making this assumption.

Though I find their model and experimental inferences compelling, the most direct test of their fundamental hypothesis would be to mutate the interface of Sst2 and Ste2. Their discussion suggests that the interface domain of Sst2 (DEP) is known as well as the corresponding binding site on Ste2. If this experiment is not viable for a specific reason, a short justification would help readers understand why it wasn't done.

Readers might find the modeling easier to follow if the authors explicitly describe the simplifications in their Carousel model, where they use a three-state G-protein cycle. It should be made clear to readers that the GDP/GTP exchange is considered synonymous with the dissociation of the G α -G β G γ complex. This assumption could be justified in three ways, a mechanistic link between nucleotide exchange and complex dissociation, if the time-scales are sufficiently separated, or if the model considers an effective rate for this transition. The authors should clarify which justification they are using.

Rebuttal letter.

Original decision letter from the editor.

Dear Alejandro,

Thank you again for submitting your work to Molecular Systems Biology. We have now heard back from three of the four referees who agreed to evaluate your manuscript. Rather than delaying further the process, I prefer to make a decision now with the available reports, given that the overall recommendations are similar. As you will see from the reports below, the referees find the topic of your study of potential interest. They raise, however, several important concerns on your work, which should be convincingly addressed in a major revision of the study.

We are happy that the reviewers found our work of potential interest. In this revised version, we have addressed all of the reviewers' concerns. Moreover, during the experiments aimed at addressing one of the reviewers' questions, we found an important issue with our own experiments. Fortunately, we have solved it satisfactorily, and together with the other changes we introduced, we have strengthened our conclusions a great deal. The paper is much better now (from the experimental side).

Below, before the point by point response to the reviewers, I explain the problems we faced, and what we did to solve them, and what we found in doing so.

The recommendations provided by the reviewers are very clear. One of the recurrent points raised by the three reviewers refers to issues related to the use of the human RGS4. In the light of the reviewers's comments it would therefore be important to perform some of the key experiments with a mutant of Sst2 that cannot interact with Ste2 (reviewer #2 provides a concrete example of such a mutant).

We answer to each reviewer this question. Here, in summary, I'd like to say that we do not agree with the idea that a mutant Sst2 unable to interact Ste2 is a good experimental avenue in this case. The reason for that is that such a mutant, as shown previously, cannot act on Gpa1 at all since it cannot reach the plasma membrane, and thus, it behaves as true Δ sst2 (Ballon et al 2006). Thus, there is not much scientific point to be made by showing experiments in such a mutant. The only potential interest would be to show that there is no extra activity performed by Sst2 (beyond binding Ste2) required for robustness to receptor abundance. In this respect, we now make a similar point by other means: we

include experiments where we “destroyed” robustness by replacing Sst2 with RGS4, and then “restored” the lost robustness by forcing RGS4 to interact with Ste2 in two different ways (more details below). This gives strong support to our model of robustness based on the coupling in a complex the GEF activity of a GPCR and the GAP activity of an RGS, as well as helps (we hope) dispel any concerns related to the use of RGS4 and to potential extra functions of Sst2 unrelated to the GAP activity.

However, to compensate for not doing that particular experiment (the mutant Sst2), we did a number of extra experiments, some in the $\Delta sst2$ strains. However, performing careful dose-response experiments, varying the abundance of Ste2 in such strains, is practically impossible. This is because we use $\Delta bar1$ strains so as to be able to control the input (α factor concentration), and the double knockout $\Delta sst2\Delta bar1$ is extremely sensitive, and it readily accumulates sterile mutations.

So sensitive that when we tested the effect of reducing Ste2 below WT abundance in a supersensitive strain, we found no dose response at all: no response in the absence of α factor or full response with even pM concentrations (the reason for testing below-WT abundances will be apparent in our answers below and in the revised version). We even found significant number of cells that executed a full response in the absence of external stimulus. We attribute this to self-stimulation due to low level expression and secretion of α factor by MATa cells, which has been documented previously. So, when we could, we tested a strain devoid of Sst2 ($P_{GAL1}\text{-SST2}$ in glucose repression conditions, Fig. EV4C).

On a more editorial level, we would kindly attract your attention to the following points:

1. The computational model should be provided in SBML format and deposited to Biomodels or an equivalent database such as JWS. Please include the accession number in a 'Data and model availability' section at the end of Materials & Methods. If it is absolutely not possible the model has to be included as computer file with appropriate documentation (submission as a zip archive).

Done.

2. As you may have noticed, we recently replaced Supplementary Information by Expanded View (EV, see examples in <http://msb.embopress.org/content/11/6/812>). In this format, a limited number of Supplementary Figures (max 5) can be integrated in the article as EV figures that are interactively collapsible/expandable and will be typeset by the publisher. In

this case, the figures should be cited as 'Figure EV1, Figure EV2" etc... in the text and their respective legends should be added to the main text after the legends of regular figures. The illustrations should be provided as separate files.

Done.

*3. For the figures that you do NOT wish to display as Expanded View figures items, they should be bundled together with their legends in a 'traditional' supplementary PDF, now called the *Appendix*. Appendix should start with a short Table of Content and the figures should be named and referred to in the main text as: "Appendix Figure S1, Appendix Figure S2" etc. See detailed instructions regarding expanded view here: <http://msb.embopress.org/authorguide#expandedview>.*

Done

4. Additional Tables/Datasets should be labeled and referred to as Table (or Dataset) EV1 etc. Table/Dataset legends can be provided in a separate tab in case of .xls files. Alternatively, you can upload a .zip file containing the Table/Dataset file and a separate README .txt file with the legend/description.

*5. We would also strongly encourage you to include the source data for figure panels that show essential data, so that readers can download these data directly from the figure. Source data files are associated to individual panels of main figures. *Numerical data* should be provided as individual .xls files (including a tab describing the data) or csv or tab-delimited text files. *For 'blots' or microscopy*, uncropped images should be submitted. For *network visualization*, Cytoscape session files, if available, can be supplied. The files should be labeled as "Source Data for Figure 1A" etc. Source Data for Expanded View and Appendix figures should be uploaded as a single ZIP file containing all the Source Data for Expanded View and Appendix content. (Additional information on source data is available in the "Guide for Authors" section at <http://msb.embopress.org/authorguide#sourcedata>).*

Done

*6. When you resubmit your manuscript, please download our CHECKLIST (http://embopress.org/sites/default/files/Resources/EP_Author_Checklist_Master.xlsx) and include the completed form in your submission. *Please note* that the Author Checklist will be published alongside the paper as part of the transparent process <http://msb.embopress.org/authorguide#transparentprocess>.*

If you feel you can satisfactorily deal with these points and those listed by the

referees, you may wish to submit a revised version of your manuscript. Please attach a covering letter giving details of the way in which you have handled each of the points raised by the referees. A revised manuscript will be once again subject to review and you probably understand that we can give you no guarantee at this stage that the eventual outcome will be favorable.

Kind regards,

Thomas

*Thomas Lemberger, PhD
Chief Editor
Molecular Systems Biology*

General explanation on the changes we have introduced to the MS.

Due to question #3 by reviewer 1, paraphrased here: “*Do the initial differences in Ste2 abundance obtained using P_{GAL1} -STE2 remain after stimulation, or are they washed away due to the induction of P_{STE2} -STE2?*”), we decided to measure Ste2 abundance over time, during the response (we had previously only measured Ste2 before stimulation).

In the new Figure 2a we show the measurements. The answer to the reviewer’s question is that it depends on the time after stimulation: at relatively short times, the difference holds, but at later times, it does not. From the data, we conclude that due to the combined effect of endocytosis and induction of P_{STE2} -STE2, after two hours the initial difference is gone.

Regarding our data in the initial submission, the Ste5 recruitment data still holds true (original Fig 2c), because we measured it at less than 5 minutes (we have data from 0.8 minutes to 6 minutes, every 50 seconds). The transcriptional data is not OK (original Figure 2b). Thus, we began a series of experiments, that involved making numerous new yeast strains to be able to actually achieve controlled changes in the abundance of Ste2.

Changes made in this new version:

- 1) We used three different strategies that together enabled us to obtain a wide range of Ste2 abundances at the plasma membrane, covering from 0.1X to 40X WT amounts (measured by direct binding using fluorescent α factor).
 - a) For large overexpression, we used the 2 micron multicopy plasmid. This allowed us to compare WT with 20-40X WT abundance
 - b) For not so large overexpression, we made mutant receptors with greatly

reduced endocytosis (we mutated 20 S/T phosphorylation sites to A in the C terminal tail of Ste2 and the 7 Ks to R (Ballon et al 2006)). This mutant shows still some endocytosis due to a minor ubiquitylation-independent pathway, but when we compare P_{STE2} -STE2 vs P_{GAL1} -STE2, in gal/raff, we obtained a 3x difference after two hours of stimulation (Fig 2E).

- c) For underexpression, we used the “GEV” strategy: we introduced the GEV chimera (Gal4DBD-EREED-VP16) pioneered by Picard years ago, to control the P_{GAL1} promoter with estradiol. In these strains, we used estradiol to cover a range from 0.1 to ~WT abundance.
- 2) Another important change we introduced is the way we control the expression of the RGS ortholog RGS4. In the original submission, we used the rather strong constitutive P_{ACT1} promoter. We realized that was a bad decision, since the endogenous RGS, Sst2, is upregulated during α factor response. Thus, we were changing two variables at the same time: the RGS and the regulation of the RGS. Now we replaced SST2 by hsRGS4 in the endogenous locus.
- 3) We now show for Ste5 recruitment and for the transcriptional reporter strains that the lost robustness in strains with hsRGS4 instead of Sst2 is recovered by forcing RGS4 binding to Ste2. We do this by either fusing the RGS domain of hsRGS4 to the DEP domain of Sst2 (as suggested by reviewer 1, question #2) or by making a Ste2-RGS4 chimera. We believe these experiments strengthen our paper greatly.
- 4) We tested the prediction of our model in a different way. We include experiments using mutant receptors that are unable to bind α factor. These receptors have been shown to act as dominant negative receptors, but the mechanism of action is not clear (see the MS for details). We reasoned that co-expression of equal amounts of DN with WT receptors decreases the maximum possible fraction of occupied receptors to 50%, but it leaves unchanged the absolute number of occupied receptors. Since we claim that the α factor pathway responds to the fraction instead of the absolute receptor occupancy, this co-expression scenario should result in 50% reporter expression at saturating α factor concentrations. Basically this is what we see. Not only that, we also show that the DN activity requires the DN receptor to be able to recruit an RGS to the plasma membrane to act on the WT receptor. We do that by fusing the DN receptor to RGS4.
- 5) We further extended the analysis of the *Carousel* model in the light of the results obtained with the newly made strains.

How have our results changed from the original submission?

- 1) We found that at the level G protein activation (Ste5 recruitment), robustness (a 5X change in Ste2 numbers) depends on the formation of the Ste2-Sst2 complex, as in the original MS.
- 2) At the transcriptional level, we found a different behavior when overexpressing than underexpressing. In both cases there was a degree of robustness. In the case of overexpression, yeast overcompensate the increased Ste2 displaying a degree of inhibition of the response. This behavior was not dependent on Ste2-Sst2 interaction or on any RGS activity. In the case of underexpression, yeast were robust in a manner that required Ste2-Sst2 interaction, just as our *Carousel* model predicted. However, that robustness was lost at high α factor stimulation. In the MS we speculate on the reasons for this behavior. Here, I would like to say that we were able to increase WT robustness using the Ste2-RGS4 fusion, which is robust even at high α factor doses.
- 3) Because of the overexpression results described in 2, we further studied our model, and found that it also predicted little or no effect of “delocalizing out of the receptor” the RGS activity when simulating increasing Ste2 abundance. The model actually predicted that delocalization of RGS was to have an effect when reducing Ste2 amounts. We have clearly missed this point in the original submission.
- 4) The experiments with the DN receptors further support the notion that the α factor pathway measures fractional occupancy, and that it does so via the “paradoxical” Ste2-Sst2 complex.

Reviewer #1:

This is a quite interesting work that presents a nice model that can explain how a G-protein response can track the fraction of GPCRs that are bound by ligand, as opposed to the total number of bound receptors. In so doing, the authors are able to explain many published experimental observations in the yeast system. The paper also contains very high quality quantitative data, as is typical of work from Dr. Colman-Lerner's lab.

Thanks you very much for this comment. We really appreciate it.

Major Points

1. From the point of view of modeling, there are some points that have been glossed over. The model nicely explains how the amount of active Galpha can be made responsive to the fraction of occupied receptors. However, in the yeast system it is beta-gamma that transmit the signal to Ste5 and Ste20, so several

other parameters must of necessity play a role, particularly the rate of recapture of beta-gamma by alpha-GDP, and the extent to which this is inhibited by the binding of beta-gamma to Ste5 and Ste20.

Thanks for this comment. To answer this point, we have extended the *Carrousel* model to include the effect of Ste5 binding to $G\beta\gamma$. This resulted in three new parameters which we obtained from published data and our own experiments. Analysis of this step indicated negligible effect of Ste5 on free $G\beta\gamma$. This seems not only due to Ste5 low abundance (500 molecules of Ste5 vs 2000 total $Ste4^{G\beta}$) but to relatively low affinity of this interaction, since simulating increasing Ste5 to 10 times the WT amount also had no effect on the DoR of free $G\beta\gamma$. We now include a section on the Appendix describing this topic.

2. On the experimental side, why didn't the authors analyze an Sst2 mutant that can no longer bind to Ste2 due to DEP-domain mutations? Also, did the authors consider attaching Sst2's DEP domain the hsRGS4?

To the first question, as explained in the general response above, we decided against testing such Sst2 mutant, with the idea that it would not add much to the behavior of a true $\Delta sst2$. To rule out that Sst2 had a function related to robustness that did not involve binding to Ste2, we followed the reviewer's advice and made the DEP-RGS4 fusion. This chimera restored the robustness at the Ste5 recruitment step lost in the RGS4 only strain. We hope that with this experiments we address this major concern.

3. How long were the cells exposed to pheromone before the readings were taken? Was it long enough that the pheromone inducibility of the wild-type Ste2 receptor could have diminished the difference between the WT vs GAL promoters?

We wish to thank the reviewer for this particular comment. As explained in the general response above, this question prompted a series of experiments that greatly strengthened the paper. The reviewer was in part correct: inducibility of Ste2 diminished the difference between WT and GAL promoters at the time we used for measuring the accumulated reporter (2 hours post stimulation), but the major determinant was endocytosis.

We hope that the new data takes care of this concern in full.

If the above points can be adequately addressed then I would be willing to change my rating for "Validity of Conclusions Drawn" from Medium to High.

Minor Points

1. *It is not completely clear that the long review of receptor theory in the introduction is completely necessary.*

We have now shortened that section of the introduction.

2. *Introduction: "and that it is this agonist-receptor complex produces the measured effect."*

Change to

"and that it is this agonist-receptor complex that produces the measured effect."

Or

"and that this agonist-receptor complex produces the measured effect."

done

3. *page 4: "which allowed to explain the different apparent affinities observed for some ligands"*

Change to

"which allowed them to explain the different apparent affinities observed for some ligands" Or

"explains the different apparent affinities observed for some ligands" Or

done

4. *page 12: "independent on the state of G"*

Change to "independent of the state of G"

done

Reviewer #2:

Overall, the paper is interesting and provides new insights into how Ste2 (and maybe other GPCRs) regulate signaling through the G protein trimer. Here are my further thoughts and comments:

The research appears to have been well designed, executed, and clearly communicated for the most part. The paper helps to explain more precisely how pheromone signaling appears to be insensitive to receptor abundance, and attributes this to Sst2's direct interaction with Ste2 by using a simplified mass-action model and an experimental verification.

The model itself is similar to previous G protein activation models (the authors reference a couple of them), but proposes to account for Sst2's localized GAP activity by maximizing the rate of Galpha-GTP hydrolysis when Galpha is associated with the receptor, as opposed to when it is not associated to the receptor. The rate assignments and simplifying assumptions in the model appear to be sound.

Thank you very much for this comment.

One of their key assumptions is that the receptor can bind to Galpha, regardless of whether it is bound to Gbeta. This has not been experimentally validated, but may be reasonable given that they define the GTP hydrolysis rate to be different based on whether Galpha is bound to receptor or not. Specifically, they define the hydrolysis rate when bound to the ligand-occupied receptor to be the same [as] when bound to the unoccupied receptor, but both of these rates are higher than when the G protein is not bound to the receptor at all. They also do a parameter analysis to show that this particular rate asymmetry is needed to satisfy both robustness to receptor abundance and DoRA.

The use of hsRGS4 as an experimental validation for the model is well motivated in the text, but their [there] are a few clarifications that are needed:

1) The result in Fig 5a makes sense, but in Fig 5b, it is not clear why higher receptor expression results in reduced signaling for the SST2 + hsRGS4 strain. It seems their intention is to show that signaling does not increase, but the fact that there is a substantial decrease in signaling, probably warrants some type of explanation.

In the new version of the MS, we show that overexpression of Ste2 leads to some degree of inhibition, and that this inhibition is not dependent on RGS activity. In the discussion we speculate for the reasons:

“... the mild inhibition observed when Ste2^{GPCR} was overexpressed. This inhibition was independent of RGS localization (Fig. 5), and it was detectable even in the absence of any RGS (Fig. EV4c). Some (but not all) previous works also show some degree of inhibition upon overexpression of Ste2 (Konopka and Jenness, 1991; Leavitt et al., 1999). This inhibition could be due to a scaffolding type effect: if G-protein activation requires a third component besides the bound receptor and the G-protein itself, then the likelihood of such a ternary complex might diminish when Ste2 is in excess. Alternatively, inhibition might be due to spatial effects: excess Ste2 might not be able to localize correctly at the signaling/polarity site, and thus, mislocalized Ste2 might titer out G-proteins.”

2) Instead of hsRGS4, why not use native Sst2 with the Ste2 docking domain disrupted? The allele, sst2(Q304N), accomplishes this (Ballon et al, 2006) and has been used in other reports.

As explained above in the general response to the reviewers, Sst2 mutants that do not bind to Ste2 are functionally speaking like Δ sst2. This is because it seems that the only means that Sst2 has to act on Gpa1^{Gα} is by binding to Ste2 (Ballon 2006). For that reason, we decided against this idea. Instead, the use of hsRGS4 does not have that complication, since it attaches to the plasma membrane independently of the receptor. Thus, its use accomplishes what we needed: a GAP activity towards Gpa1 that is not localized to the receptor. Then, to show that the reason for the lost robustness of strains with hsRGS4 instead of Sst2 was the fact that the RGS activity was not associated with the receptor, we forced the interaction between RGS4 and Ste2 in two different ways, and in both we were able to restore the robustness (see Figures 4 and 6).

3) Since only two receptor levels are shown (in Fig 5b and 5c), it is not clear what the relationship between receptor level and signaling is like in the hsRGS4 strain. Is it linear or graded? Does WT (grown in glucose) fall in between the two signaling levels given?

As explained above in the general response to the reviewers, in the new version of the paper we have scanned a wide range of receptor levels, from ~0.1X to 40X WT abundance. We've also changed the way RGS4 expression is controlled. In the new strains, we used the endogenous P_{SST2} promoter. This resulted in an expression pattern of RGS4 more similar to that of Sst2.

As we now explain in the MS, for lower than WT Ste2 abundances (the region in which robustness depends on the RGS binding to the GPCR), the relationship between receptor level and signaling in the RGS4 strains is graded: the more receptor, the more signaling.

The analysis of cell-to-cell variability in the hsRGS4 strains is interesting and consistent with what would be expected from recent reports (particularly from Dohlman's lab), but the description of the analysis is a little unclear.

Given the changes introduced, the data for this figure also changed. We have now tried to be clearer in our explanation of the analysis.

1) When reporting the variation statistic values, they do not specify at what dose the gene expression variation represents (for the SST2 strain). Since it does not change much with dose, was the value at 0nM used and then compared to the

basal receptor variation values?

Same as before, the text is new, and the data as well. We hope we explained it better now.

2) It would be informative to compare the signaling variation of the sst2del hsRGS4 strain with the regular sst2del strain, to see if variation is actually maximized by only removing the Sst2-Ste2 interaction. Some might expect variation to increase further if Sst2 is fully removed.

We agree that it would be interesting to know this. However, we feel that it is not necessary for the main message of the variability section, which was to show that the mechanism that helps the response to be robust to artificial changes in Ste2 abundance it also helps to reduce natural cell to cell variability.

In addition, as we explained above, it is really complicated to work with the double $\Delta bar1 \Delta sst2$ strains (we have to use $\Delta bar1$ to be able to control de α factor concentration properly). In this double knock out strains, the sensitivity is at least 1000-fold that of $\Delta bar1$, and, as explained above, it readily accumulates sterile mutations. To circumvent this, we used a P_{GAL1} -SST2 strategy, and maintained the strain in gal/raff. In glucose, these strains are again, extremely sensitive, what forces us to use picomolar concentrations of α factor. At these low concentrations of stimulus, ligand depletion starts to be an issue, forcing us to dilute the number of yeast, making experimentally harder to obtain enough cells to properly quantify variability.

Reviewer #3:

This manuscript describes both experimental and systems modeling approaches for understanding important properties of the yeast pheromone response pathway that are likely to have general implications for G protein coupled receptor-dependent signaling pathways. The paper focuses, in particular, on modeling the lack of variation in signaling responses as the number of pheromone receptors expressed at the cell surface is varied and on dose-response alignment, linear correlation of signaling at different stages of the pathway with the occupancy of receptors by agonist. A strong aspect of the manuscript is that the modeling leads to predictions about the system, particularly the importance of localization of the RGS protein Sst2p to receptors, that are then experimentally tested by assaying for the effects on signaling of introducing a mammalian RGS protein that is not expected to be localized to receptors. The results are important in view of the fact that the pheromone

pathway has been used as a test bed for evaluating procedures for quantitative analysis of signaling pathways. The experimental parts of the paper appear to be generally well-performed and validated using different approaches. The considerable strengths of the paper provide a strong basis for publication, however, before this can happen it seems to me that several significant issues should be addressed by the authors:

Major points:

1) A major concern is that, although the action of Sst2p plays a critical role in the model, the abundance and properties (such as on- and off-rate for receptor and/or G protein, acceleration of hydrolysis rate, etc.) of this protein do not appear to be explicitly considered in the modeling beyond a generalized effect on GTP hydrolysis. The paper concludes that stable association of Sst2p with receptors is required for the observed ratiometric signaling over a range of receptor concentrations, however, published estimates of Sst2p abundance place it at or below the abundance of even chromosomally-encoded receptors under the normal STE2 promoter. It is difficult to understand why this apparently critical component is not more explicitly considered in the modeling.

We thank the reviewer for this comment. We address this point by extending the *Carousel* model to include binding/unbinding of RGS. We comment on this extension in the text and show in Figure EV3 that the inclusion of the extra reactions does not affect the overall behavior of the model. We also include details in the Appendix.

In the text we say:

Due to the importance of this interaction, we decided to include it explicitly in the model (Fig. EV3). We found that with reasonable values for the new parameters (see Appendix), this extended *Carousel* model behaves essentially in the same way than the simplified model (Fig. EV3B-D).

With respect to the question of how can a limited number of Sst2 bind to increased number of Ste2s, we find it a reasonable concern. Our interpretation is that each Sst2 does not have to be bound “permanently” to each Ste2. The interaction could be tight but dynamic, so that one Sst2 might provide RGS activity to several Ste2 molecules.

Also, in the light of our new data, that shows that overexpression leads to a mild inhibition in signaling irrespective of the nature of the RGS, we conclude that Sst2 is not involved in ratiometric signaling at the high end of the overexpression range.

2) *Some serious concerns arise in considering the critical experiment involving expression of human RGS4:*

a) *To what extent are observed effects in the sst2 deletion strain due to the deletion vs. to the expression of RGS4, since no control without RGS4 is shown?*

In the new experiments presented in this revised version, we control for this by forcing the interaction between RGS4 and Ste2, where we show that we can recover the lost robustness. From this we conclude that the observed effect of expressing RGS4 instead of Sst2 was not due to the expression of RGS4 itself but to the inability of RGS4 to bind Ste2.

As we discuss above in the general response to the reviewers, working in a double $\Delta bar1 \Delta sst2$ at low receptor abundance was practically impossible.

We hope that the new data addresses this important concern raised by the reviewer.

RGS4 seems to be doing something to affect signaling, since its expression slightly decreases basal signaling in cells containing Sst2p in Fig. 4a, but its effect on overall dose response is not clear. Also, maximal signaling levels in Fig. 5b and Fig. 5c are decreased compared to Fig. 1b and 1c (is this significant?) Deletion of SST2 would be expected to decrease the EC50, as observed for the PGAL1-STE2 strain in Fig. 5c, simply by removing the RGS. How do we know whether this is due to loss of localization or simply to partial loss of RGS activity? The fact that increased sensitivity is not seen with the WT strain is interesting, but doesn't directly answer this question.

Thanks for this comment. The reviewer is of course right in that RGS4 resulted in decreased basal signaling, both in *SST2* and in $\Delta sst2$. That was largely due to the high level at which it was expressed in those experiments. Sst2 is not that highly expressed in unstimulated cells, and it is further induced by the pheromone pathway itself. We have now changed the P_{ACT1} with the endogenous P_{SST2} promoter.

As the reviewer points out, changes in EC50 could be due to changes in the overall RGS activity or to loss of robustness due to loss of localization. Thus, changes in the EC50 should only be compared between strains that express the same RGS.

b) *A related concern is that, contrary to what is stated in the paper, much of the RGS4 does not seem to be localized to the membrane, and thus, may not be available even in delocalized form, based on Fig. S2b.*

Thanks to the reviewer for pointing this out. We realized that the images we showed were not optimal. We have now changed that and show, we hope clearly, that the majority of RGS4-CFP is membrane localized (see Fig. EV4a).

3) *There is uncertainty about interpreting the experiments determining the number of receptors on the cell surface;*

a) Despite the addition of metabolic inhibitors, there is continued endocytosis, as is evident from Fig. 1a (bottom). Why were cells incubated with ligand for 3-5 hours at 30 degrees, which certainly exacerbates this problem? Material irreversibly trapped by endocytosis should not be included as cell surface expression of receptors available for binding.

Thanks again for pointing this out. The signal that reviewer indicates to be evidence of endocytosis is actually autofluorescence. There is no detectable endocytosis after the addition of the metabolic inhibitors.

The 3-hour incubation at 30C is necessary due to the slow binding of the fluorescent α factor (FL- α F) (Ventura 2014).

We agree with the reviewer that signal from inside the cell should not be included, and we take special care to exclude it. We have previously published work where we setup a protocol for measuring membrane associated fluorescence (Bush et al 2013).

We now show a time course of binding of FL- α F in the presence of metabolic poisons in Fig. EV1, which shows that binding increases until equilibrium and that there is no reduction after that, confirming that in this condition there is not endocytosis. The associated images also show that there is no accumulation of signal inside the cell, in contrast to the clear accumulation in cells without poison.

b) The described quantitation of binding is confusing. The main text refers to a "reference value" for WT Ste2p in glucose which apparently applies to the strains in Fig. 2, which, for some reason, contain fluorescent protein insertions at the actin locus. Why? Similar comparisons for strains in Fig. S1a (for example ACL394 vs. YAB3502) give very different numbers.

c) An additional complication in the quantitation of sites is the fact that most strains used to monitor binding of HiLyte488-tagged ligand also contain YFP fused to PRM1 or STE5 and/or CFP fused to ACT1 or hsRGS4. The contributions of these proteins to binding measurements made using the YFP cube are not discussed or, apparently controlled for.

Thanks for comments b and c. We apologize for the confusing data. For strains that express any form of CFP, its signal does not interfere with our measurement of FL- α F, since we use a YFP-like channel. In the case of the Ste5-YFP3x or P_{PRM1} -YFP containing strains, we subtract the signal from strains treated identically but without addition of FL- α F.

Strains ACL394 and YAB5302 are not similar. ACL394 has the endogenous promoter driving STE2, while YAB5302 has the PGAL1 promoter. As a result, there is very different FL- α F binding (before stimulation) but similar YFP signal after 2 hour stimulation. In any case, we do not use YAB5302 in this revised version, due to the issues explained above in the general response to the reviewers.

4) The 5-fold difference between the high and low Ste2p-expressing strains used in the manuscript is rather small compared to differences obtained by other means. There is also concern that many aspects of cell metabolism, including some possibly affecting signaling, are known to change in comparing glucose vs. raf/gal-grown strains. While measurement of levels of various components under the different conditions described in the paper is a useful first step, there are other important avenues by which the different conditions could affect signaling, such as changes in phosphorylation states of components.

We do agree that this is an important issue to have in mind. We have now greatly expanded the region of Ste2 abundances studied, and we did many of the experiments in glucose, some in gal/raff. However, we always compare strains grown in the same sugar, never glucose vs gal/raff. Thus, we think our data is internally consistent. Overall, the results seem independent of the sugar used.

5) Glucose-repressed PGAL1-STE2 shows no response to alpha-factor (strains YAB5302 and YAB5313 in Figure S1c), contrary to previous results discussed in the text.

The reviewer is correct. We do not see any measurable response to α factor in glucose-repressed P_{GAL1} -STE2. We interpret this result to indicate that there is not enough Ste2 to activate the pathway.

6) In comparing the precoupling vs. collision coupling models, one reason for disfavoring the precoupling model is that reducing the number of receptors below that of G proteins should lead to a decrease in signaling. However, decreased signaling by glucose-repressed PGAL1-STE2 is reported in this manuscript and in previous studies.

We respectfully disagree with the reviewer. We do not see any signaling in glucose-repressed PGAL1-STE2. In another study (Gehret 2012) the authors see signaling but using a multicopy 2 micron plasmid. Probably in that case there is enough leak from the PGAL1 promoter so that there is enough Ste2 to elicit a response.

In this revised MS, we explore the region below WT abundance using the GEV approach. We do see a robust response, further disfavoring the pre-coupling

model.

7) This manuscript essentially consists of two separate papers, one of them being the main text and the other being the supplementary material, which includes many important details and figures, with some duplication of points from the main text. This makes it hard to read. I'd be for including more material in the main text.

We have now tried to include as much material as possible in the main text, while minimizing repetitions in the Appendix.

8) It would be helpful if the manuscript more explicitly compared the modeling approaches and results with the extensive previous modeling of this system.

We think the MS is very large as it is now. Adding the requested comparisons would take up a considerable space.

Minor points:

1) The labeling for Figure 3b is deficient. What is the $K(RG)_{off}$ for panel b and how is it different from panel d?

Thanks for pointing this out. We have fixed this issue.

2) Graphs showing dose response curves on a log scale that include "0" points should contain breaks in the axis.

Done.

3) The manuscript a number of grammatical errors and errors in the citations that should be corrected.

Done

Reviewer #4

In this article the authors address the observation that yeast pheromone signaling is robust to variation in the abundance of receptor. Since this is inconsistent with the simplest idea of a receptor-pheromone complex leading to signaling they propose a different mechanistic model for the G-protein activation and signaling cascade. They use theory, numerical simulation and experiments to support a model in which the regulator of G protein signaling (RGS) interacts with a and is activated by the pheromone receptor. This activation of the RGS's GTPase-

activating function leads to the "activated" G-protein concentration being controlled by the ratio of receptor-pheromone complexes to unbound receptors. This ratiometric measurement makes the pheromone response robust to fluctuations in receptor abundance and rise linearly with the fraction of pheromone-bound receptors.

The work is clear in its approach and chooses to develop and test a specific mechanistic hypothesis using numerical modeling and experiments. The model is built from established molecular understanding of the pheromone signaling and G-protein signaling. They propose a coupled model with reasoned assumptions to reduce the parameter count. In particular they used micro-state reversibility and the independence of binding events to reduce the model to parameters that have been evaluated in previous literature.

In summary, this work builds a thermodynamically complete molecular model of the pheromone response system. This model is consistent with the observed robustness of pheromone signaling and they localize this robustness to the RGS-receptor interaction by numerical modeling and experimental testing. This work is interesting and worth publishing with minor changes to the writing. Specifically, the writing needs more clarity in how the hsRGS4 & SST2 expressing cells are explained based on the model. Additionally the reasoning for the experiments performed needs to be more complete (see below), potentially taking better advantage of literature as presented in the discussion

Comments

In the results section when discussing the initial experiments of supplementing Sst2 with over-expressed hsRGS4 it is necessary to comment further on why the amplitude, sensitivity, and noise suppression is still the same as WT. The current writing of the article suggests that the ratiometric measurement is contingent on making RGS activity depend on its interaction with the receptor (page 17, second paragraph). The authors need to be clearer that in this experiment, the majority of the GAP stimulation must come from receptor bound Sst2 rather than free hsRGS4 and do more to justify this argument.

Thanks for the reviewer's comment. For clarity reasons, now we do not include experiments with strains expressing both Sst2 and RGS4. We felt they shifted the focus of the study and made interpretations more complex. In any case, we agree with the reviewer that we needed to explain those results clearer.

The interpretation of their model and experiments is contingent on the fact that there is a sufficient pool of cytoplasmic Sst2 for all Ste2 to be in complex, while the free Sst2 has low GAP activity (to ensure that only receptor-bound GAPs are active). It would be prudent to discuss the appropriate literature (if available) or discuss the reasoning for making this assumption.

In the discussion:

Due to the way the RGS is encoded (as a rate of GTP hydrolysis by $G\alpha$), in the model there is always enough RGS for any receptor abundances simulated. Experimentally, this might not be case. In the PRS, before stimulation there are similar number of Sst2 and Ste2 molecules (Ghaemmaghami 2003). Thus, when Ste2 is overexpressed, it is possible that there is not enough Sst2 to form complexes with all receptors, potentially preventing the push-pull mechanism to operate. However, the interaction between Sst2 and Ste2 does not have to be stable for the system to work. If complexing is fast enough, one Sst2 might visit and act as GAP on more than one Ste2. There is no published binding rates for this interaction, but the interaction does not seem to be very tight, since Sst2-GFP fusions show a mainly cytoplasmic staining (see for example Ballon et al {Ballon 2006}).

Though I find their model and experimental inferences compelling, the most direct test of their fundamental hypothesis would be to mutate the interface of Sst2 and Ste2. Their discussion suggests that the interface domain of Sst2 (DEP) is known as well as the corresponding binding site on Ste2. If this experiment is not viable for a specific reason, a short justification would help readers understand why it wasn't done.

As explained above in the general response to the reviewers, we do not think that a mutant Sst2 that is incapable of binding Ste2 is a good experiment, since previous literature shows that such a mutant behaves essentially as a $\Delta sst2$. What we did instead is to restore robustness to strains expressing RGS4 instead of Sst2 by making a chimera with the DEP domain of Sst2 and the RGS domain of hsRGS4. We also restored robustness by directly fusing the RGS domain of RGS4 to the full length Ste2.

We hope these experiments satisfy the reviewer's concern. We added an explanation to this effect in the text as well.

Readers might find the modeling easier to follow if the authors explicitly describe the simplifications in their Carousel model, where they use a three-state G-protein cycle. It should be made clear to readers that the GDP/GTP exchange is considered synonymous with the dissociation of the $G\alpha\beta\gamma$ complex. This assumption could be justified in three ways, a mechanistic link between nucleotide exchange and complex dissociation, if the time-scales are sufficiently separated, or if the model considers an effective rate for this transition. The authors should clarify which justification they are using.

We have addressed this comment and introduced changes to simplify the explanation, as suggested.

Text relevant Results section now reads:

In this scheme, axial (up and down) reactions represent binding of the ligand L to the receptor R, radial reactions represent the coupling of R with the G protein, and angular reactions, the progression through the three-state G protein activation cycle. Note that in this cycle, we considered GDP/GTP exchange and the dissociation of the $G\alpha\beta\gamma$ trimer as a single reaction with a rate determined by the slowest reaction, the dissociation of GDP from $G\alpha$ (see Appendix).

2nd Editorial Decision

22 November 2016

Thank you again for sending us your revised manuscript. As you will see, the reviewers are now supportive and we are satisfied with the modifications made. I am pleased to inform you that your paper has been accepted for publication.

Corresponding Author Name: Alejandro Colman-Lerner

Manuscript Number: MSB-16-6910